# GENERATIVE ADVERSARIAL USER PRIVACY IN LOSSY SINGLE-SERVER INFORMATION RETRIEVAL

## ABSTRACT

We consider the problem of information retrieval from a dataset of files stored on a single server under both a user distortion and a user privacy constraint. Specifically, a user requesting a file from the dataset should be able to reconstruct the requested file with a prescribed distortion, and in addition, the identity of the requested file should be kept private from the server with a prescribed privacy level. The proposed model can be seen as an extension of the well-known concept of private information retrieval by allowing for distortion in the retrieval process and relaxing the perfect privacy requirement. We initiate the study of the tradeoff between download rate, distortion, and user privacy leakage, and show that the optimal rate-distortion-leakage tradeoff is convex and that in the limit of large file sizes this allows for a concise information-theoretical formulation in terms of mutual information. Moreover, we propose a new data-driven framework by leveraging recent advancements in generative adversarial models which allows a user to learn efficient schemes in terms of download rate from the data itself. Learning the scheme is formulated as a constrained minimax game between a user which desires to keep the identity of the requested file private and an adversary that tries to infer which file the user is interested in under a distortion constraint. In general, guaranteeing a certain privacy level leads to a higher rate-distortion tradeoff curve, and hence a sacrifice in either download rate or distortion. We evaluate the performance of the scheme on a synthetic Gaussian dataset as well as on the MNIST and CIFAR-10 datasets. For the MNIST dataset, the data-driven approach significantly outperforms a proposed general achievable scheme combining source coding with the download of multiple files, while for CIFAR-10 the performances are comparable.

## 1 INTRODUCTION

Machine learning (ML) has been recognized as a game-changer in modern information technology, and various ML techniques are increasingly being utilized for a variety of applications from intrusion detection to image classification and to recommending new movies. Efficient information retrieval (IR) from a single or several servers storing such datasets under a strict user privacy constraint has been extensively studied within the framework of private information retrieval (PIR). In PIR, first introduced by Chor *et al.* (Chor et al., 1995), a user can retrieve an arbitrary file from a dataset without disclosing any information (in an information-theoretic sense) about which file she is interested in to the servers storing the dataset. Typically, the size of the queries is much smaller than the size of a file. Hence, the efficiency of a PIR protocol is usually measured in terms of the download cost, or equivalently, the download (or PIR) rate, neglecting the upload cost of the queries. PIR has been studied extensively over the last decade, see, e.g., (Banawan & Ulukus, 2018; Freij-Hollanti et al., 2017; Kopparty et al., 2011; Sun & Jafar, 2017; Tajeddine et al., 2018; Yekhanin, 2010) and references therein.

Recently, there has been several works proposing to relax the perfect privacy condition of PIR in order to improve on the download cost, see, e.g., (Lin et al., 2020; Samy et al., 2019; Toledo et al., 2016). Inspired by this line of research, we propose to simultaneously relax both the perfect privacy condition and the perfect recovery condition, by allowing for some level of distortion in the recovery process of the requested file, in order to achieve even lower download costs (or, equivalently, higher download rates). We concentrate on the practical scenario in which the dataset is stored on a single

server. A problem formulation with arbitrary distortion and leakage functions is presented, which establishes a tri-fold tradeoff between download rate, privacy leakage to the server storing the dataset, and distortion in the recovery process for the user. We show that the optimal rate-distortion-leakage tradeoff is convex (see Lemma 1) and that it allows for a concise information-theoretical formulation in terms of mutual information in the limit of large file sizes (see Theorem 1). In the special case of full leakage to the server, the proposed formulation yields the well-known rate-distortion curve. The typical behavior of the rate-distortion-leakage tradeoff is illustrated in Fig. 1, showing that an increased level of privacy leads to a higher rate-distortion tradeoff curve, and hence a sacrifice in either download rate or distortion. A general achievable scheme combining source coding with the download of multiple files is proposed for datasets with a known distribution. Moreover, to overcome the practical limitation of unknown statistical properties of real-world datasets, we consider a data-driven approach leveraging recent advancements in generative adversarial networks (GANs) (Goodfellow et al., 2014), which allows a user to learn efficient schemes (in terms of download rate) from the data itself. In our proposed GAN-based framework, learning the scheme can be phrased as a constrained minimax game between a user which desires to keep the identity of the requested file private and a server that tries to infer which file the user is interested in, under both a user distortion and a download rate constraint. Similar to (Springenberg, 2016), where a cross-entropy loss function is used as a discriminative classifier for unlabeled or partially labeled data, the server is modeled as a discriminator in the generalized GAN framework, and also trained with cross-entropy for labeled data. We evaluate the performance of the proposed scheme on a synthetic Gaussian dataset as well as on the MNIST (Lecun et al., 1998) and CIFAR-10 (Krizhevsky, 2009) datasets. For the MNIST dataset, the data-driven approach significantly outperforms the proposed achievable scheme, while for the Gaussian dataset, where the source statistics is known, it performs close to the proposed achievable scheme using a variant of the generalized Lloyd algorithm (Lloyd, 1982; Linde et al., 1980) for the source code. For CIFAR-10, the performance of the data-driven approach is comparable to that of the proposed achievable scheme. Moreover, when the download rate is sufficiently low, it even slightly outperforms the achievable scheme.

**Related Work**

As outlined above, in this work we consider "information retrieval" in the sense of PIR, while "information retrieval" in the traditional sense used by the *information retrieval community* has a different meaning. In particular, in the traditional sense "information retrieval" refers to the problem of providing a list of documents given a query and has a wide range of applications (Baeza-Yates et al., 1999). In Wang et al. (2017), the authors proposed to iteratively optimize two well-established models of traditional information retrieval; namely generative retrieval focusing on predicting relevant documents

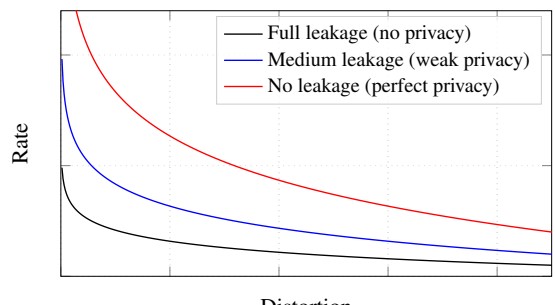

Figure 1: The rate-distortion tradeoff under different privacy levels.

given a query and discriminative retrieval focusing on predicting document relevance given a query and document pair. The resulting optimization problem is formulated as a minimax game. Due to differences in system model there is no clear connection to our proposed framework, besides the formulation as a minimax game.

Similar data-driven approaches, under the names of generative adversarial privacy (Huang et al., 2017; 2018), privacy-preserving adverisal networks (Tripathy et al., 2019), and compressive privacy GAN (Tseng & Wu, 2020), have recently been proposed for learning a privatization mechanism directly from the dataset in order to release it to the public and for generating compressed representations that retain utility while being able to withstand reconstruction attacks. A similar approach was also taken in (Blau & Michaeli, 2019) where a tri-fold tradeoff between rate, distortion, and perception in lossy image compression was established. Relaxing the perfect information-theoretical privacy condition by considering computationally-private information retrieval where the privacy requirement relies on an intractability assumption (e.g., the hardness of deciding quadratic residuosity) has been investigated in several previous works, see, e.g., (Kushilevitz & Ostrovsky, 1997; 2000; Lipmaa, 2005). Hence, given infinite computational power, the requested file index can be determined precisely. Moreover, in (Kadhe et al., 2020), it was shown that allowing for side information can also

decrease the download cost in single-server PIR. In contrast to these previous works, here we propose to relax the perfect reconstruction constraint in order to decrease the download cost. Moreover, to the best of our knowledge, addressing PIR or the extension to nonperfect privacy and recovery in the context of generative adversarial models has not been considered in the open literature so far.

## 2 PRELIMINARIES AND SYSTEM MODEL

### 2.1 NOTATION

We denote $[a] \triangleq \{1, 2, \ldots, a\}$. An arbitrary field is denoted $\mathbb{F}$, while the set of nonnegative real numbers is denoted by $\mathbb{R}_{\geq 0}$. Vectors are denoted by bold letters and sets by calligraphic uppercase letters, e.g., $\boldsymbol{x}$ and $\mathcal{X}$, respectively. We use uppercase letters for random variables (RVs) (either scalar or vector), e.g., $X$ or $\boldsymbol{X}$. For a given index set $[M]$, we write $X^{[M]}$ to represent $\{X^{(m)} \colon m \in [M]\}$. $\mathsf{E}_X[\cdot]$ and $\mathsf{E}_{P_X}[\cdot]$ denote expectation with respect to the RV $X$ and the probability mass function $P_X$, respectively. $\mathsf{H}(X)$ or $\mathsf{H}(P_X)$ represents the entropy of $X$, while $\mathsf{I}(X;Y)$ denotes the mutual information (MI) between $X$ and $Y$. The multivariate Gaussian distribution with mean $\boldsymbol{\mu}$ and covariance matrix $\Sigma$ is denoted as $\mathcal{N}(\boldsymbol{\mu}, \Sigma)$. In particular, if the entries of this distribution are mutually independent, we have $\Sigma = \sigma^2 I$, for marginal standard deviation $\sigma \in \mathbb{R}_{\geq 0}$ and $I$ denoting the identity matrix. The transpose of a vector is denoted as $(\cdot)^\intercal$.

### 2.2 SINGLE-SERVER INFORMATION RETRIEVAL

Consider a dataset containing $M$ files $\boldsymbol{X}^{(1)}, \ldots, \boldsymbol{X}^{(M)}$ stored on a single server, where each file $\boldsymbol{X}^{(m)} = (X_1^{(m)}, \ldots, X_\beta^{(m)})^\intercal$, $m \in [M]$, can be seen as a $\beta \times 1$ random vector (according to some probability distribution $P_{\boldsymbol{X}^{(m)}}$) over $\mathbb{F}^\beta$, where $\mathbb{F}$ is any field (finite or infinite). Assume that the user wishes to retrieve the $M$-th file, $M \in [M]$, where, for simplicity, $M$ is assumed to be uniformly distributed over $[M]$.[1] The formal definition of a general single-server IR scheme is as follows.

**Definition 1.** *An IR scheme $\mathscr{C}$ for a single-server storing $M$ files consists of: (i) A random strategy $\boldsymbol{S}$. (ii) A deterministic query function $f_Q$ that generates a query $\boldsymbol{Q} = f_Q(M, \boldsymbol{S})$, where query $\boldsymbol{Q}$ is sent to the server. (iii) A deterministic answer function $f_A$ that returns the answer $\boldsymbol{A} = f_A(\boldsymbol{Q}, \boldsymbol{X}^{[M]})$ back to the user. (iv) A deterministic reconstruction function $\hat{\boldsymbol{X}} \triangleq f_{\hat{X}}(\boldsymbol{A}, M, \boldsymbol{Q})$ giving an estimate of the desired file using the answer from the server together with the requested file index $M$ and the query $\boldsymbol{Q}$.*

Note that the server does not use $\boldsymbol{S}$ directly to produce $\boldsymbol{A}$ and thus using $\boldsymbol{Q}$ is sufficient in the reconstruction.

We are interested in designing an IR scheme such that both the user's *utility* and *privacy* are preserved. On the one hand, this scheme should satisfy the condition of retrievability with a distortion measure $d(\cdot, \cdot)$, i.e.,

$$\mathsf{E}_{M,\boldsymbol{Q}}\left[d(\boldsymbol{X}^{(M)}, \hat{\boldsymbol{X}})\right] \leq \mathsf{D}, \tag{1}$$

where $\mathsf{D}$ is a given distortion constraint. On the other hand, the user would like to preserve her privacy with the query function, in the sense that the server should not be able to fully determine the identity $M$ of the requested file. The server receives the query $\boldsymbol{Q}$, and the leakage is measured in terms of a leakage metric $\rho(P_{\boldsymbol{Q}|M})$. The query function should be designed such that

$$\rho(P_{\boldsymbol{Q}|M}) \leq \mathsf{L}, \tag{2}$$

where $\mathsf{L}$ is the maximum allowed leakage.

Moreover, it is worth mentioning that unlike the setting of classical PIR, where perfect retrievability is ensured for every file, i.e., $\mathsf{E}_{\boldsymbol{Q}}[d(\boldsymbol{X}^{(m)}, \hat{\boldsymbol{X}})] = 0$ for all $m \in [M]$, the distortions for different files need not be the same. In other words, it is possible to have $\mathsf{E}_{\boldsymbol{Q}}[d(\boldsymbol{X}^{(m)}, \hat{\boldsymbol{X}})] \neq \mathsf{E}_{\boldsymbol{Q}}[d(\boldsymbol{X}^{(m')}, \hat{\boldsymbol{X}})]$ for $m \neq m'$, and (1) can be expressed as

$$\mathsf{E}_{M,\boldsymbol{Q}}\left[d(\boldsymbol{X}^{(M)}, \hat{\boldsymbol{X}})\right] = \mathsf{E}_M\left[\mathsf{E}_{\boldsymbol{Q}}\left[d(\boldsymbol{X}^{(M)}, \hat{\boldsymbol{X}})\Big| M = m\right]\right] \leq \mathsf{D}.$$

---

[1]The assumption that $M$ is uniformly distributed can be lifted.

Given a query function $f_Q$ and an answer function $f_A$ of an IR scheme, we measure its efficiency in terms of the download rate (in bits per symbol) defined as

$$R(f_Q, f_A) \triangleq \frac{\mathsf{E}_{Q,A}\left[\ell_Q(A)\right]}{\beta} = \frac{1}{\beta}\sum_q P_Q(q)\,\mathsf{E}_A\left[\ell_q(A)\right], \tag{3}$$

where $\ell_q(a)$ is the length of the answer $a$ for query $q$. Further in the paper, the length $\ell_q(a)$ is a constant (i.e., independent of $a$ and $q$) that is chosen according to the desired rate.

## 2.3 PROBLEM FORMULATION

We consider a model that protects the privacy of the user against a honest-but-curious server. The IR model above guarantees that a user can retrieve an arbitrary file stored on a single server (with some distortion), while the server can only partially infer the identity of the requested file. In this work, we will study the tri-fold tradeoff between download rate, expected distortion, and privacy leakage to the server for an IR scheme.

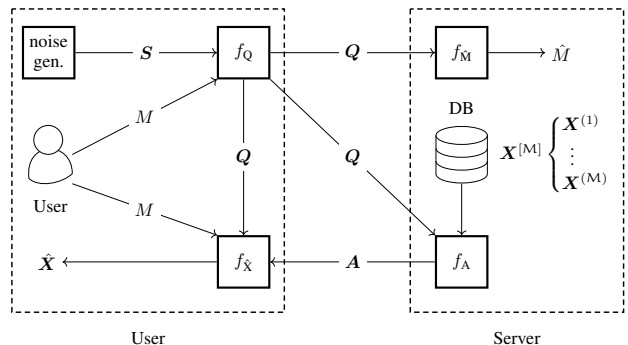

Figure 2: IR scheme for an arbitrary dataset stored on a single server.

In particular, for a given constraint D on the expected distortion $\mathsf{E}_{M,Q}\left[d(X^{(M)}, f_{\hat{X}}(A, M, Q))\right]$ for the retrieval of the requested file indexed by $M$ and for a given constraint L on the leakage to the server $\rho(P_{Q|M})$, the goal is to minimize the download rate $R(f_Q, f_A)$. This can be formulated as the constrained optimization problem

$$\min_{f_Q}\min_{f_A}\quad R(f_Q, f_A) \tag{4a}$$

$$\text{subject to}\quad \mathsf{E}\left[d(X^{(M)}, \hat{X})\right] \leq D, \quad \rho(P_{Q|M}) \leq L, \tag{4b}$$

$$\hat{X} = f_{\hat{X}}(A, M, Q), \quad Q = f_Q(M, S), \quad A = f_A(Q, X^{[M]}). \tag{4c}$$

To further capture the information leakage for training, we define $\hat{M} \triangleq f_{\hat{M}}(Q)$ to be the server's inference of the user's requested file index $M$ from the query $Q$ using a decision decoder $f_{\hat{M}}$. The proposed model is illustrated in Fig. 2. The leakage on the identity of the requested file can also be quantified by a loss function denoted as $f_{\text{Loss}}(M, \hat{M})$. Hence, the expected loss with respect to $M$ and $Q$ is

$$J(f_Q, f_{\hat{M}}) = \mathsf{E}_{M,Q}\left[f_{\text{Loss}}(M, \hat{M})\right] = \mathsf{E}_{M,Q}\left[f_{\text{Loss}}\left(M, f_{\hat{M}}(Q)\right)\right].$$

Here, the leakage metric $\rho(P_{Q|M})$ in (2) is connected to the expected loss $J(f_Q, f_{\hat{M}})$ as $\rho(P_{Q|M}) = \max_{f_{\hat{M}}} J(f_Q, f_{\hat{M}})$. Note that the main objective is to minimize the user's leakage to the server, which is equivalent to minimizing $J(f_Q, f_{\hat{M}})$, while the goal of the server is the opposite, i.e., of making the user's loss as large as possible or, equivalently, of maximizing $J(f_Q, f_{\hat{M}})$.

**Lemma 1.** *The set of achievable (feasible) rate-distortion-leakage triples* $(R, D, L)$ *from* (4) *is a convex set, for any leakage metric $\rho$ that is convex in $P_{Q|M}$.*

We remark that Lemma 1 also holds for any loss function $f_{\text{Loss}}$, since any $\rho(P_{Q|M})$ that can be expressed via $J(f_Q, f_{\hat{M}})$ is always convex in $P_{Q|M}$.

Note that if $f_{\text{Loss}}(m, \hat{M})$ is the 0-1 loss function (Nguyen & Sanner, 2013) it can be easily shown, following a similar argument as in Huang et al. (2017, Sec. 2.2), that the optimal inference strategy for the server is the MAP decoder, and that the privacy metric $\rho(P_{Q|M})$ in this case equals the server's *inference accuracy* $\Pr(M = \hat{M})$. This case is referred to as hard decision decoding. However, if $f_{\text{Loss}}(m, \hat{M})$ is the log-loss function defined as

$$f_{\text{Loss}}(m, \hat{M}) = H(M) + \log F_{\hat{M}}(m|Q), \tag{5}$$

then the leakage to the server is measured in terms of MI, i.e., $\rho(P_{\boldsymbol{Q}|M}) = \mathsf{I}(M; \boldsymbol{Q})$. This case is referred to as soft decision decoding. Moreover, the detailed derivation, which can be found in Appendix C, gives an explicit reason (see (C.12)) to use empirical categorical cross-entropy as loss function for the data-driven approach in Section 4. It can be seen that the leakage metrics $\rho$ we considered here are all convex in $P_{\boldsymbol{Q}|M}$. Thus, in the rest of paper, we will always assume that the metric $\rho(\cdot)$ is convex in its argument.

**Theorem 1.** *The download rate given in* (4) *is equal to the following information rate-distortion-leakage function defined as*

$$\mathsf{R}(\mathsf{D}, \mathsf{L}) \triangleq \min_{P_{\boldsymbol{Q}|M}, P_{\hat{\boldsymbol{X}}^{[\mathsf{M}]}|\boldsymbol{X}^{[\mathsf{M}]}, \boldsymbol{Q}} \in \mathcal{F}(\mathsf{D}, \mathsf{L})} \mathsf{I}\big(\boldsymbol{X}^{[\mathsf{M}]}; \hat{\boldsymbol{X}}^{[\mathsf{M}]} \,\big|\, \boldsymbol{Q}\big), \tag{6}$$

*as $\beta \to \infty$, where*

$$\mathcal{F}(\mathsf{D}, \mathsf{L}) \triangleq \Big\{ P_{\boldsymbol{Q}|M}, P_{\hat{\boldsymbol{X}}^{[\mathsf{M}]}|\boldsymbol{X}^{[\mathsf{M}]}, \boldsymbol{Q}} \colon \ \mathsf{E}_{M, \boldsymbol{Q}}\big[d(\boldsymbol{X}^{(M)}, \hat{\boldsymbol{X}}^{(M)})\big] \leq \mathsf{D}, \ \rho(P_{\boldsymbol{Q}|M}) \leq \mathsf{L} \Big\}.$$

The proof that the rate R is bounded from below by $\mathsf{R}(\mathsf{D}, \mathsf{L})$, for any $\beta$, is given in Appendix B. While the achievability proof is not provided, we do give the following intuition and relative technique from information theory. Note that our single-server IR scheme can be seen as a source coding problem with side information, where the server and the user act as an encoder and a decoder, respectively, and where the query can be seen as a controllable side information known to both the encoder and decoder in order to enforce the privacy condition. This is similar to the well-known Wyner-Ziv problem (Wyner & Ziv, 1976). Hence, a random-coding based scheme can be used to achieve $\mathsf{R}(\mathsf{D}, \mathsf{L})$ as $\beta \to \infty$. Further, it can be shown that the alphabet size of the designed queries can be restricted to $M + 4$, which follows from applying the Carathéodory's theorem (Cover & Thomas, 2006, Thm. 15.3.5). Note, however, that finding an analytical closed-form expression for $\mathsf{R}(\mathsf{D}, \mathsf{L})$ in our case is intractable even for independent and identically distributed (i.i.d.) files. Hence, we provide numerical results for Gaussian data in Section 5, which indicate that, by combining the achievable scheme proposed below in Section 3 with a convexifying approach, (6) is achievable.

### 2.4 GENERATIVE ADVERSARIAL APPROACH

The inference of the server can also be modeled in a generative adversarial fashion (Huang et al., 2017; 2018; Tripathy et al., 2019; Tseng & Wu, 2020). In particular, the *leakage-distortion* tradeoff for any fixed download rate constraint $\overline{\mathsf{R}}$ is formally described as follows. Consider a family of IR schemes with query generators $f_{\mathsf{Q}}$, answer functions $f_{\mathsf{A}}$, and with download rate $\mathsf{R}(f_{\mathsf{Q}}, f_{\mathsf{A}})$ at most $\overline{\mathsf{R}}$. The goal of the server is to maximize the expected loss $\mathsf{J}(f_{\mathsf{Q}}, f_{\hat{\mathsf{M}}})$ of the user by designing the decision function $f_{\hat{\mathsf{M}}}$. In contrast, the user would like to design a scheme with $f_{\mathsf{Q}}, f_{\mathsf{A}}$, and $f_{\hat{\mathsf{X}}}$ such that the download rate $\mathsf{R}(f_{\mathsf{Q}}, f_{\mathsf{A}}) \leq \overline{\mathsf{R}}$ and such that the expected loss $\mathsf{J}(f_{\mathsf{Q}}, f_{\hat{\mathsf{M}}})$ is minimized, while preserving the utility of the scheme, i.e., in the sense that the user can still retrieve the requested file with an expected distortion smaller than a prescribed D. This leads to the constrained minimax optimization problem

$$\min_{f_{\mathsf{Q}}} \max_{f_{\hat{\mathsf{M}}}} \quad \mathsf{J}(f_{\mathsf{Q}}, f_{\hat{\mathsf{M}}}) \tag{7a}$$

$$\text{subject to} \quad \mathsf{E}_{M, \boldsymbol{Q}}\Big[d(\boldsymbol{X}^{(M)}, \hat{\boldsymbol{X}})\Big] \leq \mathsf{D}, \quad \mathsf{R}(f_{\mathsf{Q}}, f_{\mathsf{A}}) \leq \overline{\mathsf{R}}, \tag{7b}$$

$$\hat{\boldsymbol{X}} = f_{\hat{\mathsf{X}}}(\boldsymbol{A}, M, \boldsymbol{Q}), \quad \boldsymbol{Q} = f_{\mathsf{Q}}(M, \boldsymbol{S}), \quad \boldsymbol{A} = f_{\mathsf{A}}(\boldsymbol{Q}, \boldsymbol{X}^{[\mathsf{M}]}). \tag{7c}$$

Note that the minimax formulation in (7a) can be written in a GAN form (Goodfellow et al., 2014) in which $f_{\hat{\mathsf{M}}}$ plays the role of the *discriminator* and $f_{\mathsf{Q}}$ plays the role of the *generator*. Thus, the machinery of GANs can be used to determine the leakage-distortion tradeoff. In doing so, (7) is first reformulated as the unconstrained optimization problem

$$\min_{f_{\mathsf{Q}}} \max_{f_{\hat{\mathsf{M}}}} \Big[ \mathsf{J}(f_{\mathsf{Q}}, f_{\hat{\mathsf{M}}}) + \eta_1 \, \mathsf{E}_{M, \boldsymbol{Q}}\big[d(\boldsymbol{X}^{(M)}, f_{\hat{\mathsf{X}}}(\boldsymbol{A}, M, \boldsymbol{Q}))\big] + \eta_2 \mathsf{R}(f_{\mathsf{Q}}, f_{\mathsf{A}}) \Big], \tag{8}$$

where $\eta_1$ and $\eta_2$ are tuning parameters, and $\boldsymbol{Q}$ and $\boldsymbol{A}$ are according to Definition 1. The minimax game in (8) with soft decision decoding, i.e., with the log-loss function in (C.11), will be the basis for training, as described below in Section 4.

## 3 ACHIEVABLE SCHEMES

In this section, we present a general achievable scheme for an arbitrary number of files $M$ and hard decision leakage (or accuracy) $L \in \{1, 1/2, 1/3, \ldots, 1/M\}$.

The construction is based on the following fact. If $L = 1$, then the problem becomes essentially the problem of lossy compression: the user explicitly tells the server what she needs to download, and the server sends the requested data compressed with some pre-agreed method. One well-known example of such compression is quantization. This scheme is designed against a maximum-likelihood decoder that provides the best inference for the server. Assume that the user wishes to retrieve the $M$-th file and to ensure leakage of $L = 1/N$, $1 \leq N \leq M$. We describe the steps of the scheme as follows.

- The user first selects a lossy source coding scheme $\mathcal{C}$ to encode together $N$ files of size $\beta$ each. Assume that the source coding scheme achieves an average compression size (in bits) of $\log_2 |\mathcal{C}|$ and expected normalized distortion $D$ (normalized by $N\beta$), respectively.
- The user's query is designed to exactly request $N$ files with indices $M_1, \ldots, M_N$ such that $M \in \{M_1, \ldots, M_N\}$, where the $N - 1$ nondesired indices are chosen uniformly at random. These $N - 1$ file indices are added in order to "trick" the server and hide the real file index $M$ of interest.
- After receiving the queries sent by the user, the server compresses all $N$ files by using the source coding scheme $\mathcal{C}$, and transmits the answer back to the user.
- The user decompresses the answer using the selected source coding scheme. The $N - 1$ nondesired reconstructed files are discarded by the user, while the desired reconstructed file is kept.

The rate of the corresponding IR scheme is $R = \log_2 |\mathcal{C}|/\beta$, while the expected distortion is $D$ and the leakage is $L$. The performance of this scheme is strongly dependent on the used source coding scheme $\mathcal{C}$. Using Lemma 1, a scheme for any leakage $1/M \leq L \leq 1$ (not only reciprocal of an integer) can be constructed.

From the general scheme above, two specific schemes can be constructed based on the source coding scheme used. The first scheme is constructed by selecting a source coding scheme based on quantization. In particular, we use a variant of the generalized Lloyd algorithm (Lloyd, 1982; Linde et al., 1980). In the sequel, we will refer to this scheme as the compression-based scheme. The second scheme, referred to in the sequel as Shannon's scheme, assumes $\beta \to \infty$ and uses the well-known rate-distortion function from information theory for the source coding scheme (Gray, 1973). In Fig. 3(a) below, together with a convexifying approach described in Appendix D.2, we plot the performance of both schemes for an i.i.d. Gaussian dataset. It is worth mentioning here that by numerically solve (6) for Gaussian data, the lower bound and the approximation values of Shannon's scheme are actually quite close, which shows that the converse bound is achievable for the i.i.d. Gaussian case.

## 4 DATA-DRIVEN APPROACH

In this section, we describe in detail our proposed data-driven framework for constructing an efficient IR scheme for downloading an arbitrary file from an arbitrary dataset stored on a single server. The four functions in Fig. 2 are represented as deep neural networks and we assume, for now, that they have already been trained.

The user wishes to retrieve the $M$-th file $\boldsymbol{X}^{(M)}$ and encodes $M$ as a one-hot $\boldsymbol{Y} = (Y_1, Y_2, \ldots, Y_M) \in \{0, 1\}^M$, where $Y_j = 1$ if $j = M$, and $Y_j = 0$ otherwise. Next, the user generates a "noise" vector $\boldsymbol{S} = (S_1, S_2, \ldots, S_M)$, where $S_1, S_2, \ldots, S_M$ are i.i.d. according to the standard Gaussian distribution $\mathcal{N}(0, 1)$. The concatenation $(S_1, S_2, \ldots, S_M, Y_1, Y_2, \ldots, Y_M)$ is the input to a deep neural network, representing the function $f_Q$, for query generation. This network produces the query $\boldsymbol{Q}$ that is sent to the server. The intuition behind this neural network is to hide the value of $M$. The server's answer is produced by concatenating the stored data $\boldsymbol{X}^{[M]}$ and the received query $\boldsymbol{Q}$ and then feed the result into a deep neural network, representing the function $f_A$, for answer construction. The deep neural network produces the answer vector $\boldsymbol{A}$ that is sent back to the user. The user then concatenates $\boldsymbol{A}$ and $(S_1, \ldots, S_M, Y_1, \ldots, Y_M)$ and feeds the result into a deep neural network for decoding, representing the function $f_{\hat{X}}$, to produce the estimate $\hat{\boldsymbol{X}}$ of the requested file $\boldsymbol{X}^{(M)}$.

On the server side, a deep neural network, representing the function $f_{\hat{M}}$, is used to guess the identity of the requested file. The input to the network is the query vector $\boldsymbol{Q}$ and the output (using softmax) is a distribution-like vector $\boldsymbol{F} = (F_1, F_2, \ldots, F_M)$, where $\sum_{j=1}^{M} F_j = 1$, $0 \le F_j \le 1$, and where $F_j$ can be interpreted as "with probability/likelihood $F_j$, the user's requested file index $M$ is equal to $j$." The server's estimate of the user requested file index is then $\hat{M} = \arg\max_{j \in [M]} F_j$.

## 4.1 LEARNING ALGORITHM

Let $\boldsymbol{X}^{(M_l)} = \left(X_1^{(M_l)}, \ldots, X_\beta^{(M_l)}\right)$, $l \in [n]$, denote the $l$-th requested file during training and $\hat{\boldsymbol{X}}_l$ the corresponding user reconstructed estimate, where $n$ is the number of training samples. Training the deep neural networks representing the functions $f_Q$, $f_A$, $f_{\hat{X}}$, and $f_{\hat{M}}$ is done following (C.12) and (8) by first fixing the download rate $R$ and then solving the minimax optimization problem

$$\min_{(f_Q, f_A, f_{\hat{X}})} \max_{f_{\hat{M}}} \frac{1}{n} \sum_{l=1}^{n} \left[ -f_{\text{XE-Loss}}(\boldsymbol{Y}_l, \boldsymbol{F}_l) + \eta \cdot d\left(\boldsymbol{X}^{(M_l)}, \hat{\boldsymbol{X}}_l\right) \right], \tag{9}$$

where $f_{\text{XE-Loss}}(\boldsymbol{Y}_l, \boldsymbol{F}_l) = -\sum_{j=1}^{M} Y_{l,j} \log F_{l,j} = -\log F_{l,M_l}$ measures the categorical cross-entropy between the vectors $\boldsymbol{Y}_l = (Y_{l,1}, \ldots, Y_{l,M})$ and $\boldsymbol{F}_l = (F_{l,1}, \ldots, F_{l,M})$. Here, $\boldsymbol{Y}_l$ and $\boldsymbol{F}_l$ is the $\boldsymbol{Y}$-vector and $\boldsymbol{F}$-vector for the $l$-th sample. The parameter $\eta$ is a tradeoff coefficient that is increased in every epoch. The solution to the minimax optimization problem in (9) is found using an iterative algorithm employing stochastic gradient decent similar to Algorithm 1 in Huang et al. (2017). The solution is found by first maximizing by the objective function of (9) to determine the optimal $f_{\hat{M}}$ for a fixed initial triple $(f_Q, f_A, f_{\hat{X}})$. Next, the optimal triple $(f_Q, f_A, f_{\hat{X}})$ is found for the given $f_{\hat{M}}$ from the previous step by minimizing the same objective function. This iterative process is continued until convergence or the maximum number of iterations is exceeded. Note that, although training is done based on a log-loss function (see (C.11)), we evaluate the performance more intuitively using accuracy, i.e., $\Pr(M = \hat{M})$, in Section 5 below.

## 5 NUMERICAL RESULTS

We demonstrate the application of our proposed data-driven approach to a synthetic Gaussian dataset and also for the MNIST and CIFAR-10 datasets, showing that guaranteeing a certain privacy level leads to a higher rate-distortion tradeoff curve, and hence a sacrifice in either download rate or distortion. We also compare the data-driven approach with both the compression-based scheme (for all datasets) and Shannon's scheme of Section 3 (for the Gaussian dataset). The Gaussian dataset consists of $M = 4$ files, each of dimension $\beta = 3$ and drawn independently according to $\mathcal{N}(\boldsymbol{\mu}^{(m)}, \sigma^2 I)$, where $\boldsymbol{\mu}^{(1)} = (-3.0, -3.0, 3.0)^\intercal$, $\boldsymbol{\mu}^{(2)} = (3.0, 3.0, 3.0)^\intercal$, $\boldsymbol{\mu}^{(3)} = (3.0, -3.0, -3.0)^\intercal$, $\boldsymbol{\mu}^{(4)} = (-3.0, 3.0, -3.0)^\intercal$, and $\sigma = 3$. For MNIST (CIFAR-10), the training set is comprised of $n = 6000$ (5000) and size $28 \times 28$ ($32 \times 32$) images for each digit (class), and testing is performed on randomly chosen images from 10000 test images. The distortion between a requested file $\boldsymbol{X}^{(m)}$ and its estimate $\hat{\boldsymbol{X}}$ is measured as the per-symbol squared error.

In Fig. 3, we plot the accuracy, or $\Pr(M = \hat{M})$, as function of per-symbol squared error distortion for different download rates $R$ for the data-driven approach (solid curves) and the compression-based scheme (dashed curves). For MNIST a symbol is a grey-scale pixel, for CIFAR-10 a pixel of each color, while for the Gaussian dataset it is a one-dimensional Gaussian RV. As a comparison, for the Gaussian dataset, we also plot the performance of Shannon's scheme (dotted curves), assuming $\beta \to \infty$, as outlined in Section 3. For Shannon's scheme, we use the *information rate*, denoted by $R_{\text{inf}}$ and defined as $H(\boldsymbol{A}|\boldsymbol{Q})/\beta$. It is well-known that the information rate is a true lower bound on the *operational* rate from (3) (cf. Cover & Thomas (2006)). Although $R_{\text{inf}}$ is a lower bound on $R$, their numerical values are close, e.g., for the Gaussian dataset and $D = 4.35$, $R = 2$ while $R_{\text{inf}} \approx 1.97$.

As expected, one can have a higher privacy level (i.e., smaller leakage) for a given distortion at the expense of a higher download rate. For the MNIST dataset, the data-driven approach significantly outperforms the compression-based scheme, while for the Gaussian dataset it performs close to the compression-based scheme using a variant of the generalized Lloyd algorithm (Lloyd, 1982; Linde et al., 1980) for the source code. This should not come as a surprise, as for the Gaussian dataset, the probabilistic model is simple and known precisely. In particular, we believe that the generalized

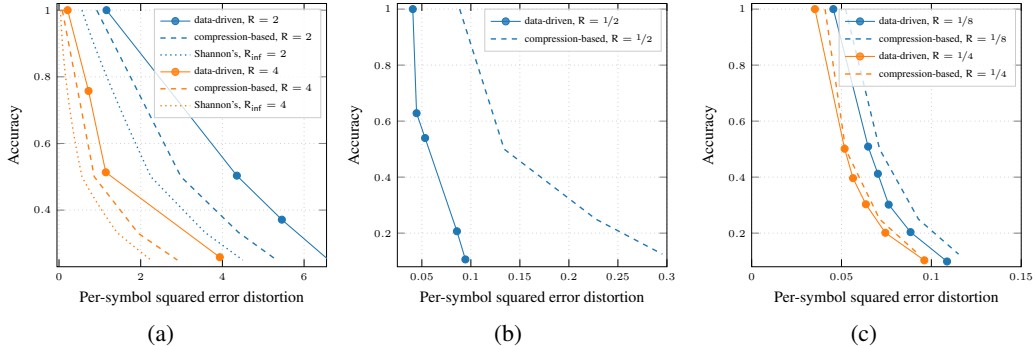

(a)                     (b)                     (c)

Figure 3: Accuracy versus per-symbol squared error distortion for both the data-driven approach and the schemes from Section 3. (a) Synthetic Gaussian dataset. (b) MNIST. (c) CIFAR-10.

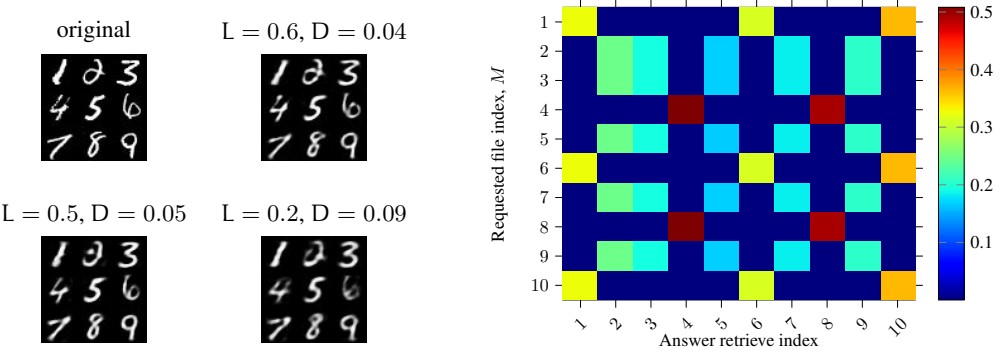

Figure 4: MNIST, $R = 1/2$ bits per pixel.

Figure 5: Heat map with CIFAR-10 for leakage $L = 0.30$, distortion $D = 0.064$, and rate $R = 1/4$.

Lloyd algorithm provides close-to-optimal compression (cf. Sabin & Gray (1986)). For MNIST we combined JPEG-like compression (including discrete-cosine transform), run-length encoding, and other entropy coding techniques for the compression-based scheme (cf. Bocharova (2010, Sec. 8.2)). However, due to a very small size of images, the constant overhead (e.g., for storing the Huffman codebook) turned out to be unacceptably high. It is a well-known phenomenon that sophisticated compression methods work well only on files of medium and large size. We thus opted for scalar quantization (see Appendix D.3 for details). Fig. 4 depicts an example of the reconstructed digits for three levels of distortion and accuracy for a download rate of $R = 1/2$ for the MNIST dataset.

For CIFAR-10 we have considered two different download rates, $R = 1/4$ and $1/8$. The compression-based scheme for CIFAR-10 is rather similar to the scheme for MNIST (see Appendix D.4). Compared to MNIST, the performance of the data-driven approach for CIFAR-10 is very close to that of the compression-based scheme for $R = 1/4$, while for $R = 1/8$ it outperforms the compression-based scheme slightly. Reasons for this might be the fact that the number of training images for CIFAR-10 is lower and that the dimension of the images is around 4 times larger than for MNIST. Hence, training is more difficult (due to the curse of dimensionality). Moreover, the images of CIFAR-10 are more structured (and with 3 separate color channels) which may make the training harder.

So far only the download cost of the proposed schemes has been considered, neglecting both the query upload cost and the cost of distributing the trained neural networks. First, the cost of distributing the trained networks can be neglected since training is usually done beforehand on a dedicated training server and can hence be seen as a one-time cost. This cost vanishes as the protocol can in principle run for a very long time serving a large number of users while the dataset grows continuously. Second, the query upload cost is in most cases much smaller than the download cost of the answers. As we will show below, this is also the case here for MNIST and CIFAR-10. For completeness, the one-time cost (in bits) of distributing the networks to the user is $465479 \times 32$, $1259135 \times 32$, and $744882 \times 32$ bits for the Gaussian, MNIST, and CIFAR-10 ($R = 1/8$) datasets, respectively,

while the corresponding one-time cost of distributing the answer generation network to the server is $532230 \times 32$, $7336888 \times 32$, and $55871908 \times 32$ ($R = 1/8$) bits, respectively.

The number of neurons of the output layer of the query network is proportional to the number of files $M$. The upload cost is at most $32 \cdot M$ (assuming 32-bits floats). For MNIST, this yields $32 \cdot 10 = 320$ bits, while the download cost is $28 \cdot 28 \cdot 1/2 = 392$ bits (assuming $R = 1/2$), which is higher. Downloading the entire dataset requires an overall communication cost of $28 \cdot 28 \cdot 8 \cdot 10 = 62720$ bits uncompressed, which is significantly higher. Moreover, the average losslessly compressed (with the LZMA algorithm (Salomon, 2007, Sec. 3.24)) sizes for different digits range from 1358 (digit 1) to 2151 (digit 8) and leads to a total average size of 19021 bits, which is again significantly higher than the download cost. Analogously, one instance of the CIFAR-10 dataset (10 files) has the size of $32 \cdot 32 \cdot 3 \cdot 8 \cdot 10 = 245760$ bits, but can be on average compressed into 197208 bits. In general, the upload cost scales linearly with the number of files and is independent of the file size, while the download cost increases with the file size. In most cases, the file size is much larger than the number of files which is the standard argument for not considering the upload cost. In the embedded table we summarize the different costs (in bits) for the Gaussian, MNIST, and CIFAR-10 datasets and compare with the cost (in bits) of downloading the entire dataset using lossless source coding.

In order to gain insight about the learned data-driven schemes, we consider CIFAR-10 and analyze the output of the *first* network of the answer generation function (in Table 3) through a heat map that reflects the contribution of each stored image in the answer, for a given requested image (further details on the construction of the heat map are given in Appendix E.1). The heat map is shown in Fig. 5 for $(L, D, R) = (0.30, 0.064, 1/4)$ and in Appendix E.1 (see Fig. 7) for $(L, D, R) = (0.40, 0.056, 1/4)$, where each row corresponds to a requested image. The answer retrieve index (the label of the $x$-axis) refers to the index of the softmax output values from the first network of the answer function, while the color reflects the actual softmax value in the sense that a warmer color indicates a higher value. As an example, consider rows 1, 6, and 10 (corresponding to $M = 1, 6$, and 10). According to the heat map, the corresponding answers are functions of the files $\boldsymbol{X}^{(1)}$, $\boldsymbol{X}^{(6)}$, and $\boldsymbol{X}^{(10)}$, meaning that the server can infer that the user is requesting one of these files, but not exactly which one, giving an accuracy of $1/3$. By looking at the remaining rows, the overall (average) accuracy becomes $L = 3/10 \cdot 1/3 + 5/10 \cdot 1/5 + 2/10 \cdot 1/2 = 3/10$. Note that the scheme in this case is similar to the compression-based scheme from Section 3 and resembles time-sharing of three different schemes with $L = 1/3, 1/5$, and $1/2$, respectively. Also, averaging the expected distortion values of the schemes for a given requested index gives exactly $D = 0.064$ (see Lemma 1).

## 6   CONCLUSION

In this work, we have studied the tradeoff between download rate, privacy leakage to the server, and reconstruction distortion at the user for single-server IR schemes. An IR scheme can be seen as an extension of the well-known

| | Download cost $(R)$ | Upload cost | Download dataset |
|---|---|---|---|
| Gaussian | $6\,(2), 12\,(4)$ | 128 | 384 |
| MNIST | $392\,(1/2)$ | 320 | 17825 |
| CIFAR-10 | $768\,(1/4)$ | 160 | 197208 |
| CIFAR-10 | $384\,(1/8)$ | 160 | 197208 |

concept of PIR by allowing for distortion in the retrieval process and relaxing the perfect privacy requirement. An information-theoretic formulation for the tradeoff in terms of mutual information has been provided in the limit of a large file size. We have shown that generative adversarial models can be successfully applied to design efficient single-server IR schemes. This is in particular beneficial if the data statistics is unknown. The main ingredient is a new optimization approach which combines GANs with additional constraints for the download rate and the desired reconstruction distortion at the user. We have shown that our proposed GAN-based data-driven approach for a fixed download rate is able to provide a tradeoff between distortion for the user and privacy leakage to the server, which is close to that of a proposed compression-based scheme where the source statistics is known. A similar tradeoff as for the Gaussian case can be observed if the proposed data-driven approach is applied to a real-world dataset like MNIST, for which it significantly outperforms the compression-based scheme. For CIFAR-10, the performance of the data-driven approach is comparable to that of the compression-based scheme.

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

# Appendix

## A    PROOF OF LEMMA 1

Assume the schemes that achieve the triples $(R_1, D_1, L_1)$ and $(R_2, D_2, L_2)$ are $\mathcal{C}_1$ and $\mathcal{C}_2$, respectively. We denote the queries of $\mathcal{C}_1$ and $\mathcal{C}_2$ as $Q_1$ and $Q_2$, respectively. From $\mathcal{C}_1$ and $\mathcal{C}_2$, we construct a new scheme $\mathcal{C}_\lambda$ that achieves the triple $(\lambda R_1 + (1-\lambda)R_2, \lambda D_1 + (1-\lambda)D_2, \lambda L_1 + (1-\lambda)L_2)$, for any $\lambda \in (0,1)$. This is exactly the definition of a convex set. The scheme $\mathcal{C}$ is as follows:

- The user generates a time-sharing RV $K$ according to the distribution

$$P_K(k) = \begin{cases} \lambda & \text{if } k = 1, \\ 1 - \lambda & \text{if } k = 2. \end{cases}$$

- Then, the user forms the query $Q = (K, Q_K)$ and sends it to the server, thus explicitly notifying which of the schemes, $\mathcal{C}_1$ or $\mathcal{C}_2$, is used.
- Finally, the server replies according to $\mathcal{C}_K$ using the query $Q_K$.

The leakage of $\mathcal{C}_\lambda$ becomes

$$\rho(P_{\boldsymbol{Q}|M}) = \rho(\lambda P_{\boldsymbol{Q}_1|M} + (1-\lambda)P_{\boldsymbol{Q}_2|M}) \overset{(a)}{\leq} \lambda\rho(P_{\boldsymbol{Q}_1|M}) + (1-\lambda)\rho(P_{\boldsymbol{Q}_2|M}) = \lambda L_1 + (1-\lambda)L_2,$$

where equality in $(a)$ can be achieved by artificially "worsening" the scheme $\mathcal{C}_\lambda$, e.g., by explicitly sending $M$ to the server in some cases (with some probability). The proof for the distortion and the rate of $\mathcal{C}$ follows in a similar manner (they are not just convex but also linear). Hence, the set of achievable rate-distortion-leakage triples constitutes a convex set.

## B    PROOF OF THEOREM 1

**Convexity.** We prove the convexity for $R(D, L)$ based on the assumption that $\rho(P_{\boldsymbol{Q}|M})$ is convex in $P_{\boldsymbol{Q}|M}$, i.e., we have

$$\begin{aligned} L_\lambda &\triangleq \rho(\lambda P_{\boldsymbol{Q}_1|M} + (1-\lambda)P_{\boldsymbol{Q}_2|M}) \\ &\leq \lambda\rho(P_{\boldsymbol{Q}_1|M}) + (1-\lambda)\rho(P_{\boldsymbol{Q}_2|M}) \leq \lambda L_1 + (1-\lambda)L_2, \quad 0 \leq \lambda \leq 1. \end{aligned}$$

Note that since the queries should be generated without knowing any realizations of the retrieved files, it is quite natural to have $\mathsf{I}(\boldsymbol{X}^{[M]}; \boldsymbol{Q}) = 0$. Thus, the objective function in the minimization of (6) can be expressed as

$$\mathsf{I}(\boldsymbol{X}^{[M]}; \hat{\boldsymbol{X}}^{[M]} \mid \boldsymbol{Q}) = \mathsf{I}(\boldsymbol{X}^{[M]}; \hat{\boldsymbol{X}}^{[M]} \mid \boldsymbol{Q}) + \underbrace{\mathsf{I}(\boldsymbol{X}^{[M]}; \boldsymbol{Q})}_{=0} = \mathsf{I}(\boldsymbol{X}^{[M]}; \hat{\boldsymbol{X}}^{[M]}, \boldsymbol{Q}).$$

Now, let the distributions $P_{\hat{\boldsymbol{X}}_1^{[M]}, \boldsymbol{Q}_1 | \boldsymbol{X}^{[M]}}$ and $P_{\hat{\boldsymbol{X}}_2^{[M]}, \boldsymbol{Q}_2 | \boldsymbol{X}^{[M]}}$ achieve $R(D_1, L_1)$ and $R(D_2, L_2)$, respectively. Given $P_{\hat{\boldsymbol{X}}_\lambda^{[M]}, \boldsymbol{Q}_\lambda | \boldsymbol{X}^{[M]}} = \lambda P_{\hat{\boldsymbol{X}}_1^{[M]}, \boldsymbol{Q}_1 | \boldsymbol{X}^{[M]}} + (1-\lambda)P_{\hat{\boldsymbol{X}}_2^{[M]}, \boldsymbol{Q}_2 | \boldsymbol{X}^{[M]}}$ for any $0 \leq \lambda \leq 1$, using the convexity of mutual information in $P_{\hat{\boldsymbol{X}}^{[M]}, \boldsymbol{Q} | \boldsymbol{X}^{[M]}}$ we obtain

$$\mathsf{I}(\boldsymbol{X}^{[M]}; \hat{\boldsymbol{X}}_\lambda^{[M]}, \boldsymbol{Q}_\lambda) \leq \lambda \mathsf{I}(\boldsymbol{X}^{[M]}; \hat{\boldsymbol{X}}_1^{[M]}, \boldsymbol{Q}_1) + (1-\lambda)\mathsf{I}(\boldsymbol{X}^{[M]}; \hat{\boldsymbol{X}}_2^{[M]}, \boldsymbol{Q}_2). \quad \text{(B.10)}$$

Further, one can also see that $P_{\boldsymbol{Q}_\lambda|M} = \lambda P_{\boldsymbol{Q}_1|M} + (1-\lambda)P_{\boldsymbol{Q}_2|M}$ and $P_{\hat{\boldsymbol{X}}_\lambda^{[M]}|\boldsymbol{X}^{[M]}} = \lambda P_{\hat{\boldsymbol{X}}_1^{[M]}|\boldsymbol{X}^{[M]}} + (1-\lambda)P_{\hat{\boldsymbol{X}}_2^{[M]}|\boldsymbol{X}^{[M]}}$. From the convexity of $\rho(\cdot)$ and the linearity of the distortion function $d(\cdot, \cdot)$, it is straightforward to see that $\lambda L_1 + (1-\lambda)L_2 \geq L_\lambda$ and $\lambda D_1 + (1-\lambda)D_2 \geq D_\lambda$. Hence, from the definition of $R(D, L)$ we get

$$\begin{aligned} R(\lambda D_1 + (1-\lambda)D_2, \lambda L_1 + (1-\lambda)L_2) &\overset{(a)}{\leq} R(D_\lambda, L_\lambda) \\ &\leq \mathsf{I}(\boldsymbol{X}^{[M]}; \hat{\boldsymbol{X}}_\lambda^{[M]}, \boldsymbol{Q}_\lambda) \\ &\overset{(b)}{\leq} \lambda R(D_1, L_1) + (1-\lambda)R(D_2, L_2). \end{aligned}$$

where $(a)$ holds since the minimization is taken over a smaller constrained set, and $(b)$ follows from (B.10). This then completes the proof of convexity of $R(D, L)$.

In the following, we show that for the case of memoryless vector sources $\boldsymbol{X}^{[M]}$, i.e., each element of $\boldsymbol{X}^{(m)}$ is chosen independently and uniformly at random from $\mathbb{F}$, the converse bound for the achievable rate $R$ is indeed $R(D, L)$.

Consider an i.i.d. sequence of length-$M$ vectors $\{\boldsymbol{X}_i^{[M]}\}_{i=1}^{\beta}$, where $\boldsymbol{X}_i^{[M]} = (X_i^{(1)}, X_i^{(2)}, \ldots, X_i^{(M)})$, $i \in [\beta]$, and each component $X_i^{(m)}$ is distributed according to $P_{X^{(m)}}$. The answer encoder $f_{\mathrm{A}}^n$ takes the input sequence $\{\boldsymbol{X}_i^{[M]}\}_{i=1}^{\beta}$ and the generated query $\boldsymbol{Q}$ to construct the codewords that are indexed by $\mathcal{A} = \{1, 2, \ldots, 2^{\beta R}\}$. The reconstruction decoder $f_{\hat{X}}^n$ outputs an estimate $\{\hat{\boldsymbol{X}}_i^{[M]}\}_{i=1}^{\beta}$ for $\hat{\boldsymbol{X}}_i^{[M]} = (\hat{X}_i^{(1)}, \hat{X}_i^{(2)}, \ldots, \hat{X}_i^{(M)})$ using the answer $\boldsymbol{A}$,[2] requested index $M$, and the generated query $\boldsymbol{Q}$ at the user side. A feasible scheme should satisfy

$$\frac{1}{\beta} \sum_{i=1}^{\beta} \mathsf{E}_{M,\boldsymbol{Q}} \left[ d(X_i^{(M)}, \hat{X}_i^{(M)}) \right] \leq D \text{ and } \rho(P_{\boldsymbol{Q}|M}) \leq L.$$

Hence, using the fact that the average code length over a source code is bounded from below by the entropy of the source, e.g., see Cover & Thomas (2006, Thm. 5.4.1), we have

$$\begin{aligned}
\beta R &\geq H(\boldsymbol{A}|\boldsymbol{Q}) \\
&\overset{(a)}{=} H(\boldsymbol{A}|\boldsymbol{Q}) - H(\boldsymbol{A}|\{\boldsymbol{X}_i^{[M]}\}_{i=1}^{\beta}, \boldsymbol{Q}) \\
&= I(\{\boldsymbol{X}_i^{[M]}\}_{i=1}^{\beta}; \boldsymbol{A} \mid \boldsymbol{Q}) \\
&\overset{(b)}{\geq} I(\{\boldsymbol{X}_i^{[M]}\}_{i=1}^{\beta}; \{\hat{\boldsymbol{X}}_i^{[M]}\}_{i=1}^{\beta} \mid \boldsymbol{Q}) \\
&\overset{(c)}{\geq} \sum_{i=1}^{\beta} H(\boldsymbol{X}_i^{[M]} \mid \boldsymbol{Q}) - \sum_{i=1}^{\beta} H(\boldsymbol{X}_i^{[M]}, \hat{\boldsymbol{X}}_i^{[M]} \mid \boldsymbol{Q}) \\
&= \sum_{i=1}^{\beta} I\left( \boldsymbol{X}_i^{[M]}; \hat{\boldsymbol{X}}_i^{[M]} \mid \boldsymbol{Q} \right), \\
&\overset{(d)}{\geq} \sum_{i=1}^{\beta} R\left( \mathsf{E}_{M,\boldsymbol{Q}} \left[ d(X_i^{(M)}, \hat{X}_i^{(M)}) \right], \rho(P_{\boldsymbol{Q}|M}) \right) \\
&\overset{(e)}{\geq} \beta \cdot R\left( \frac{1}{\beta} \sum_{i=1}^{\beta} \mathsf{E}_{M,\boldsymbol{Q}} \left[ d(X_i^{(M)}, \hat{X}_i^{(M)}) \right], \rho(P_{\boldsymbol{Q}|M}) \right) \\
&\geq \beta R(D, L),
\end{aligned}$$

where $(a)$ holds since the answer $\boldsymbol{A}$ is a function of $\{\boldsymbol{X}_i^{[M]}\}_{i=1}^{\beta}$ and $\boldsymbol{Q}$; $(b)$ follows by the data processing inequality; $(c)$ can be verified by using the chain rule of entropy and conditioning reduces entropy; $(d)$ and $(e)$ hold because $R(D, L)$ is nonincreasing and convex.

## C  Soft Decision Decoding

In contrast to hard decision decoding where the server deterministically guesses exactly one $M = m$ from the query, we can also consider a soft decision decoding rule for the server. In this case, $f_{\hat{M}}(\boldsymbol{Q})$ can be seen as a distribution over $[M]$, i.e., $f_{\hat{M}}(\boldsymbol{Q}) = F_{\hat{M}}(m|\boldsymbol{Q})$ for $m \in [M]$, where $\sum_m F_{\hat{M}}(m|\boldsymbol{Q}) = 1$. Let $f_{\mathrm{Loss}}(m, \hat{M})$ be the log-loss function defined as

$$f_{\mathrm{Loss}}(m, \hat{M}) = H(M) + \log F_{\hat{M}}(m|\boldsymbol{Q}), \tag{C.11}$$

i.e., the loss is zero when the server's guess is equally likely. Then, the expected loss function is equal to

$$J(f_{\mathrm{Q}}, f_{\hat{M}}) = H(M) - \sum_{\boldsymbol{q}} P_{\boldsymbol{Q}}(\boldsymbol{q}) \left( - \sum_m P_{M|\boldsymbol{Q}}(m|\boldsymbol{q}) \log F_{\hat{M}}(m|\boldsymbol{q}) \right)$$

---

[2]Here, without loss of generality, we assume that the reconstruction $\hat{\boldsymbol{X}}^{[M]}$ has the same dimensions as $\boldsymbol{X}^{[M]}$. However, the user can choose the desired file $\hat{\boldsymbol{X}}^{(M)}$ to retrieve.

$$= \mathsf{H}(M) - \sum_{\boldsymbol{q}} P_{\boldsymbol{Q}}(\boldsymbol{q}) \, \mathsf{H}\big(P_{M|\boldsymbol{Q}=\boldsymbol{q}}(\cdot) \, \big\| \, F_{\hat{M}}(\cdot|\boldsymbol{q})\big) \tag{C.12}$$

$$\leq \mathsf{H}(M) - \sum_{\boldsymbol{q}} P_{\boldsymbol{Q}}(\boldsymbol{q}) \, \mathsf{H}\big(P_{M|\boldsymbol{Q}=\boldsymbol{q}}(\cdot)\big) = \mathsf{H}(M) - \sum_{\boldsymbol{q}} P_{\boldsymbol{Q}}(\boldsymbol{q}) \, \mathsf{H}\big(M \, \big| \, \boldsymbol{Q} = \boldsymbol{q}\big), \tag{C.13}$$

where $\mathsf{H}\big(P_X \, \big\| \, P_Y\big)$ denotes the cross-entropy between the distributions $P_X$ and $P_Y$, and (C.13) follows since for any two distributions $P_X(\cdot)$ and $P_Y(\cdot)$, we have $\mathsf{H}\big(P_X(\cdot) \, \big\| \, P_Y(\cdot)\big) \geq \mathsf{H}\big(P_X(\cdot)\big)$. Moreover, observe that

$$\max_{f_{\hat{\mathsf{M}}}} \big(\mathsf{J}(f_{\mathsf{Q}}, f_{\hat{\mathsf{M}}})\big) \leq \mathsf{H}(M) - \mathsf{H}\big(M \, \big| \, \boldsymbol{Q}\big) = \mathsf{I}(M; \boldsymbol{Q}),$$

and from (C.13), it follows that equality holds if $P_{M|\boldsymbol{Q}=\boldsymbol{q}}(\cdot) = F_{\hat{M}}(\cdot|\boldsymbol{q})$. Therefore, under the log-loss function, the leakage to the server is measured in terms of MI, i.e., $\rho(P_{\boldsymbol{Q}|M}) = \mathsf{I}(M; \boldsymbol{Q})$, and the user wishes to minimize the MI leakage.

# D ACHIEVABLE SCHEMES FROM SECTION 3

## D.1 COMPRESSION-BASED SCHEME FOR GAUSSIAN DATA

As a reference for synthetic Gaussian data (see Fig. 3(a)), we used the compression-based scheme outlined in Section 3. As explained in Section 3, the scheme reduces to quantization of random vectors drawn from a Gaussian multinomial distribution. We used the generalized Lloyd algorithm, also known as the Linde-Buzo-Gray (or LBG) algorithm (Linde et al., 1980). Note that this algorithm is closely related to $k$-means clustering in unsupervised learning (Levrard, 2018). The authors argue in Linde et al. (1980) that although their approach is not guaranteed to provide the best quantizer, its results are nearly optimal for a wide class of distributions.

We briefly explain the steps of finding the quantization vectors (or levels) below and refer the interested reader to Linde et al. (1980) for further details. When quantizing a Gaussian vector into $r$ bits, the number of quantization vectors is $k = 2^r$.

1. First, generate a large sample of $n$ vectors $\{\boldsymbol{x}_i\}_{i=1}^n$ from the desired multinomial Gaussian distribution.

2. Then, select an initial set of quantization vectors $\{\boldsymbol{q}_j\}_{j=1}^k$.

3. Next, continue iterating as follows:

   (a) Each sample vector $\boldsymbol{x}_i$ is quantized to (approximated by) its closest quantization vector (its "nearest neighbor") $\boldsymbol{q}_{j_i^*}$ where

   $$j_i^* = \arg\min_j d(\boldsymbol{x}_i, \boldsymbol{q}_j).$$

   This gives a mean quantization error (or distortion) among all the sample vectors of

   $$\mathsf{D} = \frac{1}{n} \sum_{i=1}^n d(\boldsymbol{x}_i, \boldsymbol{q}_{j_i^*}).$$

   (b) Each quantization vector is updated to the mean of the sample vectors as follows:

   $$\boldsymbol{q}_j \leftarrow \frac{1}{|\mathcal{S}_j|} \sum_{\boldsymbol{x} \in \mathcal{S}_j} \boldsymbol{x},$$

   where the set of sample vectors that are quantized to $\boldsymbol{q}_j$ is denoted by $\mathcal{S}_j$ ("neighborhood" of $\boldsymbol{q}_j$).

4. The algorithm stops when the mean quantization error between two consecutive iterations changes less than a predefined threshold.

Since the mean quantization error (or distortion) $\mathsf{D}$ is nonnegative and nonincreasing between iterations, the algorithm is guaranteed to converge. Since it is a randomized algorithm, we ran it

multiple times and chose the quantizer that produced the smallest mean quantization error. As a final remark, we remind the reader that our goal here is not to find the best quantization possible, but only to have a good enough approximation.

Additionally, after the avhievable points are calculated, we make sure they all satisfy the convexity property (cf. Lemma 1). More precisely, if the triples $(R_1, D_1, L_1)$, $(R_2, D_2, L_2)$, and $(\lambda R_1 + (1 - \lambda)R_2, D_3, \lambda L_1 + (1 - \lambda)L_2)$ are achievable for some $0 < \lambda < 1$, we update

$$D_3 \leftarrow \min(D_3, \lambda D_1 + (1 - \lambda)D_2).$$

## D.2 Shannon's Scheme for Gaussian Data

This section briefly outlines the Shannon's scheme for Gaussian data (plotted in Fig. 3(a)). Following its description in Section 3, the server compresses $N$, $N \in [M]$, independent files, each consisting of $\beta$ independent Gaussian RVs, together. It is known from the rate-distortion theory for multi-dimensional sources Gray (1973), as the file size $\beta \to \infty$, that a download rate $R = N R_G(D)$ is achievable with distortion $D$. Here, $R_G(D) = \max\{\frac{1}{2}\log(\sigma^2/D), 0\}$ for a Gaussian RV distributed according to $\mathcal{N}(\mu, \sigma^2)$ (Cover & Thomas, 2006, Ch. 10). Similar to Section D.1, we further use a convexifying approach to obtain an achievable scheme for any hard decision leakage (or accuracy) $1/M \leq L \leq 1$. In the following we brief describe the scheme. First, we select two accuracy values, say $L_1$ and $L_2$, $L_1, L_2 \in \{1, 1/2, \ldots, 1/M\}$. Applying the Gaussian rate-distortion function of $R_G(D) = \frac{1}{2}\log(\sigma^2/D)$, $D \in (0, \sigma^2]$, to the previously outlined scheme, one can obtain two achievable distortions $D_G(R_1, L_1) = \sigma^2 \cdot 2^{-2R_1 L_1}$ and $D_G(R_1, L_1) = \sigma^2 \cdot 2^{-2R_2 L_2}$. Next, the desired accuracy $L \in [L_1, L_2]$ is selected to be $L = \lambda L_1 + (1 - \lambda)L_2$, and the goal is to determine the minimum possible linear combination of $D_G(R_1, L_1)$ and $D_G(R_2, L_2)$ subject to $R = \lambda R_1 + (1 - \lambda)R_2$, i.e.,

$$\min_{\substack{L_1^{-1}, L_2^{-1} \in [M] \\ L = \lambda L_1 + (1-\lambda)L_2}} \min_{R = \lambda R_1 + (1-\lambda)R_2} \left[ \lambda D_G(R_1, L_1) + (1 - \lambda)D_G(R_2, L_2) \right]. \tag{D.14}$$

This then gives the performance of the synthetic Gaussian dataset in Fig. 3(a). As a final remark, we also numerically evaluate the values of $R(D, L)$ for the Gaussian dataset. For instance, for $D = 1.875$ and $L = 0.6$, we obtain $R(D, L) \approx 2.0345$, while (D.14) gives 1.875 for $(R, L) = (2.0, 0.6)$, confirming that the performance of our proposed Shannon's scheme is optimal.

## D.3 Compression-Based Scheme for the MNIST Dataset

The scheme is based on the ideas of Section 3 and reduces to choosing a lossy compression method for one image. First, we applied a grayscale compression similar to the instruments used by the JPEG standard. We refer the interested reader to (Bocharova, 2010, Sec. 8.2) for more details and other techniques for image compression. The main ingredient of the JPEG standard is a two-dimensional discrete cosine transform. In the standard, an image is split into $8 \times 8$ blocks and the transform is applied to each block independently. Since the images from the MNIST dataset are of size $28 \times 28$, we applied the transform to $7 \times 7$ blocks. In this way, an image is split into 16 blocks. Next, the resulting values were quantized by a uniform scalar quantizer. For each block, the top-left coefficients play the most important role as they grasp the low-frequency contents of the block. Thus, they are assigned more bits from the available pool of bits (defined by the desired rate). Finally, these quantized coefficients were encoded with a combination of run-length encoding and a two-dimensional Huffman code.

Since the MNIST images have a small size, the entropy coding techniques do not provide any compression benefits. For example, the usually ignored overhead of storing the Huffman coding table requires too many bits in our case. Also, the run-length encoder performs poorly as there are not many repetitive values. In fact, it even increases the size of an MNIST image on many occasions. Therefore, we turned to a much simpler scalar quantization of the original grayscale images where each image is split into blocks. For each block, the average of values of its pixels was quantized and stored. This method also allows for easier control of the desired rate.

### D.4 COMPRESSION-BASED SCHEME FOR THE CIFAR-10 DATASET

The compression-based scheme for CIFAR-10 is rather similar to the scheme for MNIST. The only difference being a standard image preprocessing stage that is very common when compressing color images. First, the image is transformed from red, green, and blue channel representation to luminance-chrominance representation, consisting of brightness, hue, and saturation channels. Next, the hue and saturation channels are decimated by a factor of 2. In other words, four neighboring pixels that form a $2 \times 2$ block are described by four values of brightness, one value of hue, and one value of saturation. These channels are further quantized in a similar manner as for MNIST.

## E LEARNING ALGORITHM FOR THE DATA-DRIVEN APPROACH

---

**Algorithm 1:** Training algorithm for generative adversarial user privacy

---

**Input** : Number of training samples $n$, training samples $(\boldsymbol{X}_1^{[M]}, \ldots, \boldsymbol{X}_n^{[M]})$, number of training iterations $T$, size of minibatch for stochastic gradient decent $b$, and initial tuning parameter $\eta_{\text{initial}}$

**Output :** $f_Q$, $f_A$, $f_{\hat{X}}$, and $f_{\hat{M}}$

1  $t \leftarrow 1$
2  $\eta \leftarrow \eta_{\text{initial}}$
3  Initialize the neural networks representing $f_Q$, $f_A$, $f_{\hat{X}}$, and $f_{\hat{M}}$
4  **while** $t \leq T$ **do**
5     **for** $m \in [M]$ **do**
6       Generate a minibatch $\mathcal{B}_m \subseteq [n]$ of indices corresponding to the $m$-th file, $|\mathcal{B}_m| = b$
7       Generate $b$ noise samples $\boldsymbol{s}_{\mathcal{B}_m}$ from the distribution $\mathcal{N}(\boldsymbol{0}, I)$
8       **for** $b \in \mathcal{B}_m$ **do**
9         $\boldsymbol{q}_{m,b} \leftarrow f_Q(m, \boldsymbol{s}_b)$
10      **end**
11    **end**
12    Update $f_{\hat{M}}$ by descending its stochastic gradient:
13    $\sum_{m=1}^{M} \sum_{b \in \mathcal{B}_m} \frac{1}{b \times M} \log F_m(\boldsymbol{q}_{m,b})$
14    **for** $m \in [M]$ **do**
15      Generate a minibatch $\mathcal{B}_m \subseteq [n]$ of indices corresponding to the $m$-th file, $|\mathcal{B}_m| = b$
16      Generate $b$ noise samples $\boldsymbol{s}_{\mathcal{B}_m}$ from the distribution $\mathcal{N}(\boldsymbol{0}, I)$
17      Generate a minibatch of $b$ training samples $\{\boldsymbol{X}_b^{(m)} : b \in \mathcal{B}_m\}$
18      **for** $b \in \mathcal{B}_m$ **do**
19        $\boldsymbol{q}_{m,b} \leftarrow f_Q(m, \boldsymbol{s}_b)$
20      **end**
21    **end**
22    **for** $m \in [M]$ **do**
23      **for** $b \in \mathcal{B}_m$ **do**
24        $\boldsymbol{a}_{m,b} \leftarrow f_A(\boldsymbol{q}_{m,b}, \boldsymbol{X}_b^{([M])})$
25      **end**
26    **end**
27    Update $f_Q$, $f_A$, and $f_{\hat{X}}$ by descending its stochastic gradient:
28    $\sum_{m=1}^{M} \sum_{b \in \mathcal{B}_m} \frac{1}{b \times M} \left[ \log F_m(\boldsymbol{q}_{m,b}) + \eta \cdot d\left(\boldsymbol{X}_b^{(m)}, f_{\hat{X}}(\boldsymbol{a}_{m,b}, m, \boldsymbol{q}_{m,b})\right) \right]$
29    Update the tuning parameter $\eta$
30    $t \leftarrow t + 1$
31 **end**
32 **return** $f_Q$, $f_A$, $f_{\hat{X}}$, and $f_{\hat{M}}$

---

As elaborated in Section 4.1, learning is done by first fixing the download rate $R$ and then solving the minimax optimization problem in (9). Here, the rate is computed based on the dimension of the answer network's output and the corresponding quantizer levels like Blau & Michaeli (2019); Mentzer et al. (2018). In Algorithm 1 we outline the detailed training algorithm, which is similar to Algorithm 1 in Huang et al. (2017). The learning rate is selected empirically for each accuracy

and distortion level with the RMSprop optimizer, and the number of iterations is $T = 100000$. The number of training samples, denoted by $n$, for the Gaussian case is set to a sufficiently large number (samples can easily be generated from a Gaussian RV) and to 60000 (6000 for each digit) and 50000 (5000 for each class) for the MNIST and CIFAR-10 datasets, respectively.

The architectures of the deep neural networks representing the functions $f_Q$, $f_A$, $f_{\hat{X}}$, and $f_{\hat{M}}$ used for training for the synthetic Gaussian dataset are detailed in Table 1. In the special case of accuracy $L = 1$, i.e., full leakage, the problem reduces to the classical rate-distortion problem, and to improve performance, we only considered an encoder (for compression of the requested file), corresponding to $f_A$, and a decoder (for decompression at the user side), corresponding to $f_{\hat{X}}$. Both the encoder and the decoder are represented as deep neural networks and trained in the classical way (Kingma & Welling, 2014).

Table 1: Neural network architectures used for training for the synthetic Gaussian dataset. FC stands for fully connected layer. Size is the number of neurons or input size. As activation function we used the scaled exponential linear unit (SeLU) (Klambauer et al., 2017).

| Query generator, $f_Q$ | | Answer generation, $f_A$ | |
|---|---|---|---|
| Size | Layer | Size | Layer |
| 8 | Input | $4 + 3 \times 4 \times 1$ | Input |
| 8 | FC, SeLU | 16 | Flatten |
| 8 | FC, SeLU | 256 | FC, SeLU |
| 8 | FC, SeLU | 256 | FC, SeLU |
| 8 | FC, SeLU | 256 | FC, SeLU |
| 4 | FC, SeLU | 256 | FC, SeLU |
| | | 256 | FC, SeLU |
| | | 256 | FC, SeLU |
| | | 256 | FC, SeLU |
| | | 256 | FC, SeLU |
| | | 256 | FC, SeLU |
| | | Answer dim | FC, tanh |

| Decoder, $f_{\hat{X}}$ | | Adversary, $f_{\hat{M}}$ | |
|---|---|---|---|
| Size | Layer | Size | Layer |
| Answer dim $+8$ | Input | 4 | Input |
| 256 | FC, SeLU | 64 | FC, SeLU |
| 256 | FC, SeLU | 64 | FC, SeLU |
| 256 | FC, SeLU | 64 | FC, SeLU |
| 256 | FC, SeLU | 64 | FC, SeLU |
| 256 | FC, SeLU | 64 | FC, SeLU |
| 256 | FC, SeLU | 64 | FC, SeLU |
| 256 | FC, SeLU | 4 | FC, softmax |
| 256 | FC, SeLU | | |
| 3 | FC | | |

For the MNIST dataset there are $M = 10$ files (there are 10 digits) and each file is of size $\beta = 28 \times 28 = 784$ symbols, or $784 \times 8 = 6272$ bits (each picture is of size $28 \times 28$ pixels and each pixel is of size 8 bits). The architectures of the deep neural networks representing the functions $f_Q$, $f_A$, $f_{\hat{X}}$, and $f_{\hat{M}}$ in this case are given in Table 2.

For the CIFAR-10 dataset there are also $M = 10$ files (there are 10 classes of files) and each file is of size $\beta = 32 \times 32 = 1024$ symbols, or $1024 \times 8 \times 3 = 24576$ bits (each picture is of size $32 \times 32$ pixels and each pixel is of size 8 bits for each of the 3 color channels). The architectures of the deep neural networks representing the functions $f_Q$, $f_A$, $f_{\hat{X}}$, and $f_{\hat{M}}$ in this case are given in Table 3. Note that in contrast to the Gaussian dataset and MNIST, in Table 3 we list descriptions of three neural networks for the answer function $f_A$, labeled as part 1 to 3. To produce the final answer, the output of

the first network is fed to the third network together with the output from the second network and the queries (which are fed into the fourth layer of the network). The input to the first network is the queries, while the input to the second network is the database. The second network takes the database as input and extracts its features. As for the Gaussian case, training for both MNIST and CIFAR-10 can be simplified for L = 1 as described above.

Table 2: Neural network architectures used for training for the MNIST dataset. The input pixel value is rescaled between $-1.0$ and $1.0$. FC stands for a fully connected layer. Conv and ConvT stand for a convolutional and a transposed convolutional layer, respectively, where "st" is shorthand for stride. Size is the number of neurons or input size. As activation function we used SeLU and hyperbolic tangent (tanh).

Query generator, $f_Q$

| Size | Layer |
|---|---|
| 20 | Input |
| 20 | FC, SeLU |
| 20 | FC, SeLU |
| 20 | FC, SeLU |
| 10 | FC, SeLU |

Answer generation, $f_A$

| Size | Layer |
|---|---|
| $10 + 10 \times 28 \times 28$ | Input |
| $26 \times 26 \times 8$ | Conv, SeLU |
| $12 \times 12 \times 8$ | Conv (st = 2), SeLU |
| $10 \times 10 \times 16$ | Conv, SeLU |
| $4 \times 4 \times 16$ | Conv (st = 2), SeLU |
| 256 | Flatten |
| 2560 | Concatenate 10 parallel outputs of previous layers |
| 1024 | FC, SeLU |
| 1024 | FC, SeLU |
| 1024 | FC, SeLU |
| 1024 | FC, SeLU |
| 512 | FC, SeLU |
| 512 | FC, SeLU |
| 512 | FC, SeLU |
| 512 | FC, SeLU |
| 392 | FC , tanh |

Decoder, $f_{\hat{X}}$

| Size | Layer |
|---|---|
| 412 | Input |
| 512 | FC, SeLU |
| 512 | FC, SeLU |
| 512 | FC, SeLU |
| 512 | FC, SeLU |
| $4 \times 4 \times 32$ | Unflatten |
| $11 \times 11 \times 64$ | ConvT (st = 2), SeLU |
| $25 \times 25 \times 128$ | ConvT (st = 2), SeLU |
| $28 \times 28 \times 1$ | ConvT (st = 1), tanh |

Adversary, $f_{\hat{M}}$

| Size | Layer |
|---|---|
| 10 | Input |
| 64 | FC, SeLU |
| 64 | FC, SeLU |
| 64 | FC, SeLU |
| 64 | FC, SeLU |
| 64 | FC, SeLU |
| 64 | FC, SeLU |
| 10 | FC, softmax |

In order to assess the quality of the learning, we tabulate below an estimate of the inference accuracy of a MAP adversary for $M$, denoted by $\mathsf{L_{MAP}}$, operating directly on the queries from the query network and also an estimate of the entropy of the answer $A$ from the answer network for the

Table 3: Neural network architectures used for training for the Cifar-10 dataset. The input pixel value is rescaled between $-1.0$ and $1.0$. FC stands for a fully connected layer. Conv and ConvT stand for a convolutional and a transposed convolutional layer, respectively, where "st" is shorthand for stride. Size is the number of neurons or input size. As activation function we used SeLU, sigmoid, and hyperbolic tangent (tanh).

| Query generator, $f_Q$ | | Answer generation part 1, $f_A$ | |
|---|---|---|---|
| Size | Layer | Size | Layer |
| 20 | Input | 5 | Input |
| 20 | FC, SeLU | 5 | FC, SeLU |
| 20 | FC, SeLU | 7 | FC, SeLU |
| 9 | FC, SeLU | 9 | FC, SeLU |
| 7 | FC, SeLU | 10 | FC, softmax |
| 5 | FC, SeLU | | |

Answer generation part 2, $f_A$

| Size | Layer |
|---|---|
| $10 \times 32 \times 32 \times 3$ | Input |
| $10 \times 30 \times 30 \times 8$ | Conv, SeLU |
| $10 \times 14 \times 14 \times 8$ | Conv (st = 2), SeLU |
| $10 \times 12 \times 12 \times 16$ | Conv, SeLU |
| $2304 \times 10$ | Concatenate 10 parallel outputs of previous layers |

Answer generation part 3, $f_A$

| Size | Layer |
|---|---|
| $10 + 2304 \times 10$ | Input |
| $2304 \times 10$ | Broadcast and elementwise multiply |
| 23040 | Flatten |
| 5 | Input |
| 2304 | FC, SeLU |
| 1024 | FC, SeLU |
| Answer dim | FC, sigmoid |

| Decoder, $f_{\hat{X}}$ | | Adversary, $f_{\hat{M}}$ | |
|---|---|---|---|
| Size | Layer | Size | Layer |
| Answer dim + 20 | Input | 5 | Input |
| 512 | FC, SeLU | 64 | FC, SeLU |
| 512 | FC, SeLU | 64 | FC, SeLU |
| 512 | FC, SeLU | 64 | FC, SeLU |
| $4 \times 4 \times 32$ | Unflatten | 64 | FC, SeLU |
| $8 \times 8 \times 16$ | ConvT (st = 2), SeLU | 64 | FC, SeLU |
| $16 \times 16 \times 8$ | ConvT (st = 2), SeLU | 64 | FC, SeLU |
| $32 \times 32 \times 3$ | ConvT (st = 2), tanh | 10 | FC, softmax |

Gaussian dataset for different levels of per-symbol squared error distortion D. The estimates have been found by Monte-Carlo simulation of the networks, i.e., by repeatedly generating queries and answers and estimating the probability density functions of the random variables $Q|M$ and $A$. For comparison we also show the inference accuracy L of the adversary network in the third line. As can be seen by comparing the second and third lines, the inference accuracy of the adversary network is close to the inference accuracy of a MAP adversary. Likewise, the entropy $H(A)$ is close of $R \cdot \beta$, which gives an indication that the performance of neural networks adversary is close to an optimal bound. For CIFAR-10 with accuracy L = 0.40 and distortion D = 0.056, the MAP accuracy is $L_{MAP} = 0.41$, which shows that also for CIFAR-10 the adversary is trained quite well.

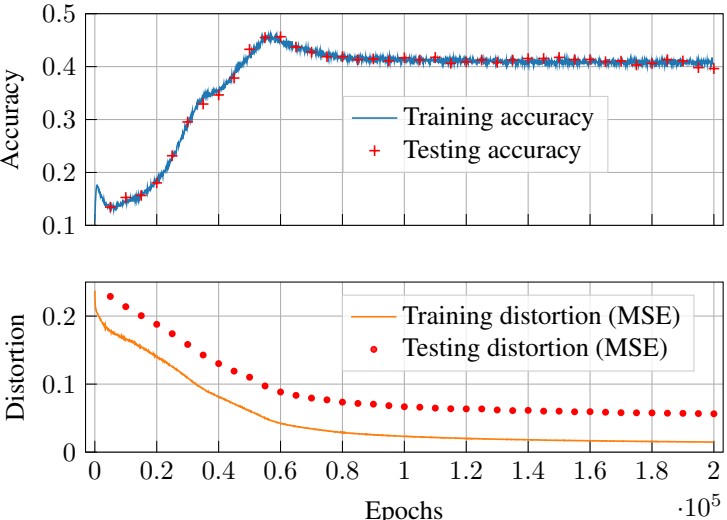

Figure 6: Learning curves with CIFAR-10 for leakage L = 0.40, distortion D = 0.056, and rate R = 1/4.

| Gaussian data set and R = 2 | | | | |
|---|---|---|---|---|
| D | 6.60 | 5.46 | 4.35 | 1.16 |
| $L_{MAP}$ | 0.26 | 0.38 | 0.50 | 1.0 |
| L | 0.25 | 0.37 | 0.50 | 1.0 |
| H($\boldsymbol{A}$) | 5.83 | 5.85 | 5.90 | 5.89 |

| Gaussian data set and R = 4 | | | | |
|---|---|---|---|---|
| D | 3.93 | 1.15 | 0.72 | 0.21 |
| $L_{MAP}$ | 0.27 | 0.53 | 0.76 | 1.0 |
| L | 0.26 | 0.51 | 0.76 | 1.0 |
| H($\boldsymbol{A}$) | 11.82 | 11.56 | 11.45 | 10.86 |

To investigate the well-known overfitting problem in ML (Goodfellow et al., 2016), we show some learning curves with the CIFAR-10 dataset (for leakage L = 0.40, distortion D = 0.056, and rate R = 1/4) in Fig. 6. The training distortion and training adversary accuracy curves are averaged over a sliding window of length 100 and then sampled every 100 epoch. The testing distortion and testing adversary accuracy curves are sampled every 5000 epoch. If overfitting happened, then the testing distortion would increase while the training distortion would decrease with the number of epochs. This does not happen in our case. In fact, the testing distortion curve is decreasing and close to the training distortion curve. Hence, the level of overfitting is negligible. Moreover, training and testing accuracy values are close.

The mode collapse issue in GANs is well-known (Salimans et al., 2016; Goodfellow, 2017). Mode collapse means that the generator finds some weak point of the discriminator and keeps producing this mode of output to trick the discriminator. This may happen when the generator tries to fool the discriminator trained on a real dataset and causes problems because a GAN model is typically required to have diversity in its output. Techniques such as minibatch discrimination (Salimans et al., 2016) can be applied to prevent mode collapse. However, note that in our step-up mode collapse is less relevant (and may even be beneficial in some cases) as we in general do not need to have diversity in the query generation. In fact, a scheme for the MNIST (or CIFAR-10) dataset producing a deterministic query is a valid scheme with MAP adversary accuracy $L_{MAP} = 1/10$, while a scheme giving a one-to-one correspondence between $M$ and $\boldsymbol{Q} \mid M$ is also valid and gives MAP adversary accuracy $L_{MAP} = 1$. In order to investigate mode collapse in more detail for intermediate values of the accuracy, we have estimated the variance $\mathsf{Var}\left[\boldsymbol{Q} \mid M\right]$ for CIFAR-10 for leakage L = 0.40, distortion D = 0.056, and rate R = 1/4. The estimated values are from $10^6$ samples and are tabulated below. As can be seen from the table, the query network generates queries with some diversity. Moreover, from the MAP accuracy values $L_{MAP}$ above (for CIFAR-10 with (L, D, R) = (0.40, 0.056, 1/4) it is 0.41), it follows that the discriminator is trained quite well and do not appear to suffer from degeneration in the query generation training due to mode collapse.

Query variances for a given requested file index for CIFAR-10 with $(\mathsf{L}, \mathsf{D}, \mathsf{R}) = (0.40, 0.056, 1/4)$.

| Requested file index $M = m$ | Var $[\boldsymbol{Q}\|M = m]$ |
|:---:|:---:|
| 1 | $(0.000, 0.000, 0.081, 0.036, 0.001)$ |
| 2 | $(0.049, 0.007, 0.024, 0.108, 0.000)$ |
| 3 | $(0.041, 0.040, 0.090, 0.004, 0.079)$ |
| 4 | $(0.000, 0.000, 0.081, 0.035, 0.001)$ |
| 5 | $(0.027, 0.041, 0.000, 0.023, 0.001)$ |
| 6 | $(0.027, 0.041, 0.000, 0.023, 0.001)$ |
| 7 | $(0.027, 0.041, 0.000, 0.023, 0.001)$ |
| 8 | $(0.000, 0.000, 0.080, 0.035, 0.001)$ |
| 9 | $(0.041, 0.040, 0.091, 0.005, 0.081)$ |
| 10 | $(0.040, 0.039, 0.091, 0.005, 0.081)$ |

### E.1 HEAT MAP REPRESENTATION OF THE ANSWER

In this subsection, we provide additional details on the heat map representation of the output of the *first* network of the answer generation function for the learned data-driven schemes for CIFAR-10. As mentioned in Section 5, the heat map reflects the contribution of each stored image in the answer, for a given requested image (corresponding to a given row of the heat map). Denote by $\delta^{(1)}, \ldots, \delta^{(M)}$ the outputs from the $M$ neurons following the softmax activation functions of the first network of the answer generation function. The second network of the answer generation function extracts the features of the database, denoted by $\boldsymbol{Z}^{(1)}, \ldots, \boldsymbol{Z}^{(M)}$, which are fed as input to the third network of the answer generation function together with $\delta^{(1)}, \ldots, \delta^{(M)}$ and the queries (which are fed into the fourth layer). In the first layer (of the third network), the vector $(\delta^{(1)}, \ldots, \delta^{(M)})$ is multiplied elementwise with the feature vector $(\boldsymbol{Z}^{(1)}, \ldots, \boldsymbol{Z}^{(M)})$, producing the vector $(\delta^{(1)}\boldsymbol{Z}^{(1)}, \ldots, \delta^{(M)}\boldsymbol{Z}^{(M)})$, which is subsequently combined with the queries in the fifth layer in order to produce the final answer. Now, if $\delta^{(m)}$, for some $m \in [M]$, is close to zero, then the answer will not depend much on the file indexed by $m$, and it follows that the heat map indeed reflects which files contribute the most to the generated answer.

In Fig. 7, the heat map for $(\mathsf{L}, \mathsf{D}, \mathsf{R}) = (0.40, 0.056, 1/4)$ is shown. As an example, consider rows $1$, $4$, and $8$ (corresponding to $M = 1$, $4$, and $8$). According to the heat map, the corresponding answers are functions of the files $\boldsymbol{X}^{(1)}$, $\boldsymbol{X}^{(4)}$, and $\boldsymbol{X}^{(8)}$, meaning that the server can infer that the user is requesting one of these files, but not exactly which one, giving an accuracy of $1/3$. For $M = 2$ (corresponding to the second row), however, the answer depends only on the second file, meaning that the server can infer that the user is requesting the second file, giving an accuracy of $1$. By looking at the remaining rows, the overall (average) accuracy becomes $\mathsf{L} = 3/10 \cdot 1/3 + 3/10 \cdot 1/3 + 3/10 \cdot 1/3 + 1/10 \cdot 1 = 4/10$.

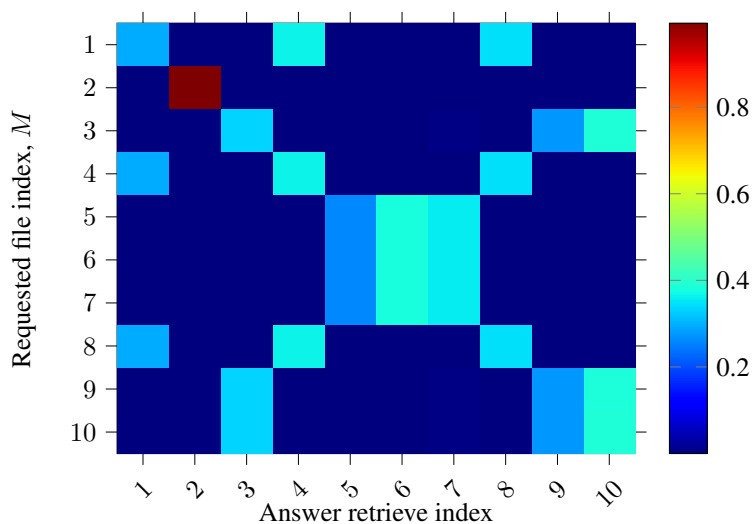

Figure 7: Heat map with CIFAR-10 for leakage L = 0.40, distortion D = 0.056, and rate R = 1/4.

