# OpenReview forum: "Generative Adversarial User Privacy in Lossy Single-Server Information Retrieval"
_ICLR.cc/2021/Conference — Reject_

### Official Review · AnonReviewer3 · 2020-10-27
**this paper extend PIR with data distortion and provide a GAN based scheme**

**Rating:** 6
**Confidence:** 1

**Review:**

Summary:
This paper studies single-server private information retrieval with user data distortion constraint. In particular, the authors formulate the problem as a minimization of download rate under distortion and privacy constraints. They show the achievable set is convex. They also provide a information-theoretic formulation of the trade-off. Based on this framework, the authors propose to parameterize query function and file estimation function by neural networks and formulate it as a two-player game. Empirically, the authors demonstrate the effectiveness on the Gaussian dataset and MNIST dataset.

Pros: 1. The trade-off between distortion, privacy, the download rate is indeed an interesting problem.


Cons: 1. The idea of using GAN for information retrieval is not new. It would be better if the authors could discuss the contributions compared to previous works like IRGAN 2017, other than adding privacy and distortion constraint.

Questions: 1. Since the authors apply the data-driven approach, I am wondering how does the overfitting problem/mode-collapse affect the trade-off.

Reasons for score: I am not an expert in information retrieval, so I could not tell the signification of the contributions. Overall the paper is well written.

---

> ### Author Response · Authors · 2020-11-17
> **Response to AnonReviewer3**
>
> We would first like to thank the reviewer for useful comments on our submitted paper. We have marked all corresponding changes in the paper in *green* color.
>
>  **Previous works:**
>
> The IRGAN work from 2017 mentioned by the reviewer is on information retrieval in the traditional sense used by the *information retrieval community* [2], while we consider information retrieval with the underlying aspect of information-theoretic privacy, i.e., that the database is not able to completely determine the identity of the requested file as some part is kept private. Further, we consider reconstruction with some user distortion, a notion not present in the classical formulation of information retrieval. In particular,  the classical formulation of information retrieval is to provide a (ranked) list of documents given a query, a formulation which is also adopted in the IRGAN paper. In contrast, our goal is to privately retrieve a file, while minimizing information leakage to the database, the resulting distortion at the user, and the download rate of the file from the database to the user. However, as a similarity, both our work and the IRGAN work employ generative models in order to optimize the retrieval scheme. In order to better highlight our contribution with respect to existing work, we have updated the related work section on page 2, where we also discuss the contribution of our work compared to the IRGAN paper. Please see the text highlighted in green.
>
> [2] R. Baeza-Yates, B. Ribeiro-Neto, *et al.*, Modern Information Retrieval, vol. 463, ACM Press, New York, 1999.
>
>
> **Overfitting/mode collapse questions:**
>
> We believe that overfitting/mode collapse is not a major concern in our work. To investigate the overfitting problem, we show some learning curves with the CIFAR-$10$ dataset (for leakage $L=0.40$, distortion $D=0.056$, and rate $R=1/4$)  in Figure 6 on page 21 in the appendix. The training distortion and training adversary accuracy curves are averaged over a sliding window of length $100$ and then sampled every $100$ epoch. The testing distortion and testing adversary accuracy curves are sampled every $5000$ epoch. If overfitting happened, then the testing distortion would increase while the training distortion would decrease with the number of epochs.  This does not happen in our case. In fact, the testing distortion curve is decreasing and close to the training distortion curve. Hence, the level of overfitting is negligible. Moreover, training and testing accuracy values are close.  Our idea is to keep the model simple to prevent overfitting. There are however techniques such as residual block or attention module that can be applied in order to increase model capability while avoiding overfitting. The main focus of this paper is an end-to-end trainable information retrieval framework, and thus such enhancements have not been investigated in depth. See also updated text on page 21 highlighted in green.
>
> The mode collapse issue in GANs is well-known [3]. Mode collapse means that the generator finds some weak point of the discriminator and keeps producing this mode of output to trick the discriminator. This may happen when the generator tries to fool the discriminator trained on a real dataset and causes problems because a GAN model is typically required to have diversity in its output. Techniques such as minibatch discrimination [3] can be applied to prevent mode collapse. However, note that in our step-up mode collapse is less relevant (and may even be beneficial in some cases) as we in general do not need to have diversity in the query generation. In fact, a scheme for the MNIST (or CIFAR-$10$) dataset producing a deterministic query is a valid scheme with MAP adversary accuracy $L_{\mathsf{MAP}}=1/10$, while a scheme giving a one-to-one correspondence between $M$ and $\mathbf{Q} \mid M$ is also valid and gives MAP adversary accuracy $L_{\mathsf{MAP}}=1$. In order to investigate mode collapse in more detail for intermediate values of the accuracy,  we have estimated the variance of $\mathbf{Q}\mid M$ for CIFAR-$10$ for leakage $L=0.40$, distortion $D=0.056$, and rate $R= 1/4$. The estimated values are from $10^6$ samples and are tabulated on page 22 in the paper. As can be seen from the table, the query network generates queries with some diversity. Moreover, from the MAP accuracy values $L_{\mathsf{MAP}}$ tabulated on page 21 (for CIFAR-$10$ with $(L,D,R)=(0.40,0.056,1/4)$ it is $0.41$), it follows that the discriminator is trained quite well and do not appear to suffer from degeneration in the query generation training due to mode collapse. See also updated text on pages 21-22 highlighted in green.
>
> [3] T. Salimans, I. J. Goodfellow, *et al.*, ````"Improved techniques for training GANs,'' in *Proc. Advances in Neural Information Processing Systems (NIPS)*, 2016.

---

### Official Review · AnonReviewer1 · 2020-10-28
**feasible and sound approach for private information retreival**

**Rating:** 6
**Confidence:** 3

**Review:**

Summary:
The paper aims at taking a new approach towards the problem of private information retrieval. The proposed method relies on the interplay of the three parameters: distortion (utility), leakage (privacy) and the download rate/cost. They try to decrease the download cost, by sacrificing some utility (lossy compression through GANs), which is an interesting and seemingly novel take on the problem. Their method needs to solve two optimization problems: the first one assumes a fixed utility and privacy, and minimizes the download rate, the second one is a minimax loss that assumes a fixed rate and trades off privacy for utility. They then propose three practical methods based on this, the first two are not applicable on all cases, the last one, however, is data driven and can be applied to a wider range of problems.

pros:
+The method offers a nice trade-off between the three dimensions of cost, utility and privacy.
+The proposed method is evaluated both theoretically and experimentally which helps verify its veracity.

cons:
- This is more of a question: why are there only two datasets, the synthesized one and MNIST? why not bigger datasets? would the proposed methods also work on larger images? I assume one reason would be because information theoretic bounds are much harder to enforce in high dimension cases. Is that the case? If so, how can it be addressed?

- Also, in Section 5, paragraph third paragraph, there was a sentence which is a bit ambiguous to me: "However, due to a very small size of images, the overhead (e.g., for storing the Huffman codebook) turned out to be unacceptably high."I did not completely understand what the problem is. Is it the case that because the images are small, the codebook becomes really large? Would this get worse with larger images?


[I am not at all familiar with information retrieval and the work surrounding it, so I am not entirely confident in my review and I might update it based on the review of expert reviewers later on].

---

> ### Author Response · Authors · 2020-11-17
> **Response to AnonReviewer1**
>
>
> We would first like to thank the reviewer for useful comments on our submitted paper. We have marked all corresponding changes in the paper in blue color.
>
> **Question for more datasets:**
>
> The proposed data-driven approach is general and can be applied to any dataset. We have initially applied it to two datasets, a synthetic Gaussian dataset and MNIST. In the updated manuscript (see Figure 3(c) on page 8) we show results also for CIFAR-$10$ (for rates $R= 1/4$ and $1/8$) in order to show its broader applicability. As can be seen from the results, the data-driven approach is still competitive compared to the compression-based scheme. For $R=1/8$ the data-driven approach even outperforms the compression-based scheme slightly. The architectures of the corresponding deep neural networks are given in Table 3 in the appendix. Please see text highlighted in blue for these updates. The compression-based scheme for CIFAR-$10$ is rather similar to the scheme for MNIST. The only difference being a standard image preprocessing stage that is very common when compressing color images. First, the image is transformed from red, green, and blue channel representation to luminance-chrominance representation, consisting of brightness, hue, and saturation channels. Next, the hue and saturation channels are decimated by a factor of $2$. In other words, four neighboring pixels which form a $2 \times 2$ block are described by four values of brightness, one value of hue, and one value of saturation. These channels are further quantized in a  similar manner as for MNIST. A description of the compression-based scheme for CIFAR-$10$ has been added to a new Appendix D.4. Compared to MNIST, the performance of the data-driven approach for CIFAR-$10$ is very close to that of the compression-based scheme. Reasons for this might be the fact that the number of training images for CIFAR-$10$ is lower and that the dimension of the images is around $4$ times larger than for MNIST. Hence, training is more difficult (due to the curse of dimensionality). Moreover, the images of CIFAR-$10$ are more structured (and with $3$ separate color channels) which may make the training harder. We would also like to remark here that when almost doubling the model size of the answer generation and decoder networks, the distortion can be further reduced by approximately $6$% for $R=1/4$ with $L=0.3$.
>
> Establishing the information-theoretic optimal tradeoff curve by solving the optimization problem of Theorem 1 is infeasible for a  high-dimensional dataset of big images. Even for MNIST this is infeasible. Moreover, optimizing the compression-based scheme for a high-dimensional dataset is difficult as finding an appropriate image compression scheme is a research topic in its own. However,  based in the insight we have gained so far for MNIST and CIFAR-$10$ we believe that the data-driven approach will also provide good results for much bigger datasets of much bigger images, as training in general will improve with the size of the dataset.
>
>
> **Comment on the third paragraph in Section 5:**
>
> The point we were trying to make here is that the overhead (e.g., for storing the Huffman codebook) is (almost) constant and its proportion to the size of the whole compressed file is diminishing for larger files. It is a well-known phenomenon of compression theory that sophisticated compression methods work well only on files of medium and large size. However, the images in the MNIST dataset are too small for a conventional image compression methods to give good compression ratios. We have added a bit more on this issue (marked in blue) in the third paragraph of Section 5.

---

### Official Review · AnonReviewer2 · 2020-10-29
**Some technical statements needs to be formalized**

**Rating:** 5
**Confidence:** 4

**Review:**

This paper considers and formulates a generalized version of the private information retrieval (PIR) problem, where a user aims to retrieve one of $M$ files from a dataset, but wants to keep the index $M$ private. Unlike the basic version studied in prior works, the problem formulated in the paper allows non-zero privacy leakage and distortion and aims to trade these two quantities as well as the download communication rate.  A data-driven approach is proposed to find PIR schemes, and a more conventional approach is presented as a benchmark.

Topic: Overall the main focus of this work is on optimization and information theory and is less relevant to DL, so conferences like ICML or ISIT might be a slightly better match.

Technical comments: Several statements in the technical formulation needs to be revised for correctness and readability. For example,
1. Lemma1 is proved by allowing the user to share some additional information to the server, which should be prohibited, otherwise, the user can simply deliver $M$ to the server and making the privacy requirement trivial. One possible fix is to send that information as part of the query and to fix related statements such as how the convexity of matrix $\rho$ should be defined.

2. There are some missing conditions in the formulations of optimization problems such as (4b), where additional constraints such as Markov chain requirements are needed for the result to be non-zero (or non-trivial).

Presentation: Regarding the plots in Figure (3), it might be good to add comments and provide intuition on why the data-driven approach is better in one setting and worse in the other.


Post Rebuttal:

I'm totally fine with the topic, which is a very minor issue as mentioned earlier. My main concern was regarding the overselling of some presented results.

From the technical side, Lemma 1 is a common approach in information theory/channel coding/privacy that one can linearly combine several designs by randomly sampling them, which was presented in the abstract as one of the main results. I expected something beyond a simple application, such as determining an interesting condition where this approach can be used, but the provided statement does not seem to hold after revision. As long as one can append additional messages to the query, the domain of Q could change completely, and the convexity of P_{Q|M}, interpreted exactly as stated, will not provide any guarantee. The condition actually needed by the authors is essentially random sampling does not hurt privacy, making the statement "random sampling can be applied when it can be applied", which is a bit trivial.

I carefully double-checked the provided proof steps and found that an additional assumption was needed in the very last step, which further requires something like the metric \rho is invariant under permutation of Q, for which I didn't find in the paper, maybe I missed it. Adding this assumption totally makes sense to me, but requiring convexity+invariant under permutation is a bit over-complication, instead of just directly requiring random sampling does not hurt privacy. The theory part of the paper should be presented in the simplest possible way, without unnecessary complication.

The authors do have a novel contribution in terms of experiments. But the contribution and novelty in the formulation are rather limited given that the considered components have been studied either separately or jointly in cited works.

For the experiments: I have a similar concern as mentioned by Reviewer 1, which is about the generalizability issue. A natural explanation of the presented results could be that the hyperparameters in the implemented solution are tweaked for datasets like MNIST or something similar. It is not clear if it would fail for more general datasets similar to synthetic Gaussian.

I've read the authors' responses to review 1 and the revised paper. Overall, there is no more direct evidence of generalizability. But this issue itself is fine as it could happen in many other works, so I won't consider it as my main concern.

Besides, the author has adequately addressed the issues of the missing conditions in the Theorems.

Conclusively, my recommendation remains the same for the above reasons.

---

> ### Author Response · Authors · 2020-11-16
> **Response to AnonReviewer2**
>
> We would first like to thank the reviewer for useful comments on our submitted paper. We have marked all corresponding changes in the paper in *purple* color.
>
> **Topic concern:**
>
> The reviewer mentions that the main focus in this paper is on optimization theory and information theory and that ISIT or ICML might be a slightly better match. However, we believe that this work is also interesting for the audience of ICLR as learning is a key ingredient of the data-driven approach. Furthermore, in the revised manuscript, we also provide a novel heat map representation of the output from the first network of the answer generation function in order to provide insight about the learned data-driven schemes and to emphasize the learning representation aspects of the paper. The heat map reflects the contribution of each stored file in the answer, for a given requested file. Heat maps for CIFAR-$10$ are provided in Figures 5 and 7 and a description is given in Section 5 with further details in Appendix E.1.
>
> Moreover, by looking through the accepted papers from last year's ICLR we have found several papers that might be considered as being in the same "category", i.e., papers that apply ML techniques to a theoretical computer science problem in order to find better "schemes". See, e.g., the work on applying deep learning techniques in order to find good graph matchings (https://openreview.net/forum?id=rJgBd2NYPH), the work on SAT solvers (https://openreview.net/forum?id=HJMC_iA5tm), and the work on sparse signal recovery (or compressed sensing) (https://openreview.net/forum?id=B1xVTjCqKQ). Compressed sensing has mainly been studied in signal processing, statistics, and information theory communities. In the mentioned work, ML techniques are used both to the design sensing schemes and to speed up signal recovery.
>
> **Comments for Lemma 1:**
>
> We agree that in the proof of Lemma 1, the user shares additional information with the server; more precisely, which scheme, $\mathcal C_1$ or $\mathcal C_2$, will be used. Our implication was exactly as the reviewer suggested: this should be sent as a part of the query. We have updated the proof of Lemma 1 accordingly by explicitly employing the leakage metric, also showing that the convexity property still holds.
>
> **Problem formulations:**
>
> In the optimization problems, the requirements on $\mathbf Q$ and $\mathbf A$ (Markov properties) were implied, according to Definition 1. We have changed the optimization problem formulations by stating explicit conditions for recovery, query and answers in (4c) and (7c), respectively.
>
> **Plots in Figure 3:**
>
> For the MNIST  dataset, the data-driven approach significantly outperforms the compression-based scheme, while for the Gaussian dataset it performs close to the compression-based scheme using a variant of the generalized Lloyd algorithm for the source code.
> This is somehow expected, as for the Gaussian dataset, the probabilistic model is simple and known precisely. In particular, we believe that the generalized Lloyd algorithm provides close-to-optimal compression (cf. [1]). On the other hand, for the MNIST dataset, the scalar quantization is simplistic, while the data-driven approach is able to learn a better solution.
>
> [1] N. Sabin and R. Gray, "Global convergence and empirical consistency of the generalized Lloyd algorithm," *IEEE Transactions on Information Theory*, vol. 32, no. 2, pp. 148-155, Mar. 1986.

---

### Decision · Program_Chairs · 2021-01-07
**Final Decision**

**Decision:**

Reject

**Comment:**

The paper got mixed ratings. However, keeping in mind the low confidence of some of the reviewers, the paper needed an additional look. The AC himself went over the paper. The paper presents an interesting formalism for private information retrieval. As reviewers have pointed out the formalism is based on several existing ideas on utility privacy tradeoff.
The use of GANs for enforcing privacy is also not new. The rebuttal did not convince some of the reviewers about novelty which seems reasonable given the area and literature in it.

Overall, the paper needs to consolidate all ideas of Adversarial training for privacy and compare and contrast with the proposed approach to make it compelling for publication.